# DNA methylation-based classification of sinonasal tumors

The diagnosis of sinonasal tumors is challenging due to a heterogeneous spectrum of various differential diagnoses as well as poorly defined, disputed entities such as sinonasal undifferentiated carcinomas (SNUCs). In this study, we apply a machine learning algorithm based on DNA methylation patterns to classify sinonasal tumors with clinical-grade reliability. We further show that sinonasal tumors with SNUC morphology are not as undifferentiated as their current terminology suggests but rather reassigned to four distinct molecular classes defined by epigenetic, mutational and proteomic profiles. This includes two classes with neuroendocrine differentiation, characterized by *IDH2* or *SMARCA4/ARID1A* mutations with an overall favorable clinical course, one class composed of highly aggressive SMARCB1-deficient carcinomas and another class with tumors that represent potentially previously misclassified adenoid cystic carcinomas. Our findings can aid in improving the diagnostic classification of sinonasal tumors and could help to change the current perception of SNUCs.

Although tumors of the sinonasal region only account for a small fraction of head and neck tumors, they encompass a diverse spectrum of epithelial, mesenchymal and neuroectodermal neoplasms[1]. The complexity of these tumors presents a major challenge for histopathological diagnosis, even for trained head and neck pathologists[2]. In fact, tumors of the sinonasal region have been reported to show the highest rate of conflicting diagnoses among all head and neck tumors[3].

Sinonasal undifferentiated carcinomas (SNUC) represent an especially challenging diagnosis. SNUCs are aggressive carcinomas that lack a definite lineage-specific differentiation[4]. For diagnostic evaluation, a variety of other entities have to be excluded, such as poorly differentiated carcinomas or high-grade olfactory neuroblastomas. Histologically, SNUCs by definition lack squamous or glandular differentiation but may show subtle neuroendocrine features and thus may focally resemble neuroendocrine carcinomas[5–7]. In recent years, molecular analyses of SNUCs have revealed a high rate of *IDH2* mutations or alterations of the switch/sucrose non-fermentable (SWI/SNF) complex leading to SMARCB1 or SMARCA4 deficiency[8–11]. These distinct molecular patterns as well as the occasional morphological and immunohistochemical resemblance to neuroendocrine carcinoma are challenging the current definition of SNUC as a single entity.

DNA methylation is an epigenetic modification of the DNA which regulates gene expression. It plays a significant role in the differentiation of different cell types and it has been shown that DNA methylation patterns are highly tissue-specific[12]. Although epigenetic alterations represent one of the hallmarks of cancer development, the global DNA methylation signature of tumor cells is thought to contain substantial information about the cell of origin, making DNA methylation an ideal tool for tumor classification[13]. From a technical perspective, methylated DNA is highly robust (in contrast to alternative molecules such as RNA), enabling the retrospective analysis of formalin-fixed and paraffin embedded (FFPE) samples, almost irrespective of sample age. Using this approach, significant cohorts of even exceedingly rare tumors can be assembled. For these reasons, DNA methylation has shown promising results in the classification of a growing number of malignancies[14–18]. DNA methylation profiling of olfactory neuroblastomas and a cohort of sinonasal carcinomas showed that *IDH2* mutated and *SMARCB1* deficient carcinomas likely represent epigenetically distinct classes[10,19]. Furthermore, it has been suggested that *IDH2* mutated neuroendocrine carcinomas and *IDH2* mutated SNUCs may represent the same entity, due to their epigenetic similarity[10,20].

For this study, we collected a cohort of 395 sinonasal tumors and relevant differential diagnoses encompassing 18 different

✉e-mail: philipp.jurmeister@med.uni-muenchen.de

histologically defined entities as well as normal sinonasal control tissue to elucidate the epigenetic landscape of these tumors. Within this dataset we identified highly robust DNA methylation-based tumor classes. By further integrating mutational profiling and mass spectrometry-based proteomics, we provide sound evidence that tumors with SNUC morphology consist of four distinct epigenetic subclasses which are supported by different driver mutations landscape and protein expression profiles. Furthermore, we provide a machine learning-based algorithm for reliable classification of diagnostic samples which may improve the histopathological diagnosis of challenging cases.

## Results

### Identification of DNA methylation-based sinonasal tumor classes

To test if DNA methylation-based tumor classification for SNUCs was applicable, we obtained a cohort of 429 high quality DNA methylation profiles of sinonasal tumors and normal tissue. A t-distributed stochastic neighbor embedding (t-SNE) dimensionality reduction and unsupervised clustering of the 20,000 most variable CpG sites was used to assess the optimal number of classes and for best partition. The distribution of the CpG sites selected for class separation showed a uniform chromosomal distribution and did not show any enrichment in the functionally relevant promotor regions when compared to the overall array design (Supplementary Fig. 1). The clustering algorithm identified 34 cases as noise or singularity points that did not correspond to a stable cluster, including a relatively high number of cases

that were histologically classified as neuroendocrine carcinomas (11/24) or SNUCs (15/84). Noise data points were excluded, resulting in a final reference set comprising of 395 samples, covering 18 tumor entities as defined in the WHO Classification of Head and Neck Tumors as well as normal sinonasal tissue[4]. The workflow for the compilation of the reference set is also summarized in Supplementary Fig. 2.

The t-SNE dimensionality reduction of the final reference set is shown in Fig. 1. A total of 18 distinct and stable epigenetic classes were identified (Supplementary Data 1). We did not observe any batch effects related to possible confounding factors (Supplementary Fig. 3). Iterative random down-sampling with correlation analysis of the t-SNE coordinates indicated a high stability of the classes with a median correlation coefficient of 0.992 (Range: 0.945 to 0.999; Supplementary Fig. 4). 14 classes were equivalent to their conventional histopathological classification as defined in the WHO classification. The remaining four DNA methylation classes included 133 tumors from a spectrum of different histological entities. Notably, all 69 SNUC samples of the reference set were among these 133 tumors. These four SNUC DNA methylation classes were further molecularly characterized (see results below) and based on these findings were assigned the provisional names NEC-like IDH2, SMARCB1, ACC and NEC-like SMARCA4/ARID1A. Summary copy number plots derived from DNA methylation data for all classes are shown in Supplementary Fig. 5.

### Reassessment of SNUC classes

To further evaluate tumor specimens that were assigned to the four SNUC classes defined by distinct DNA methylation profiles, we

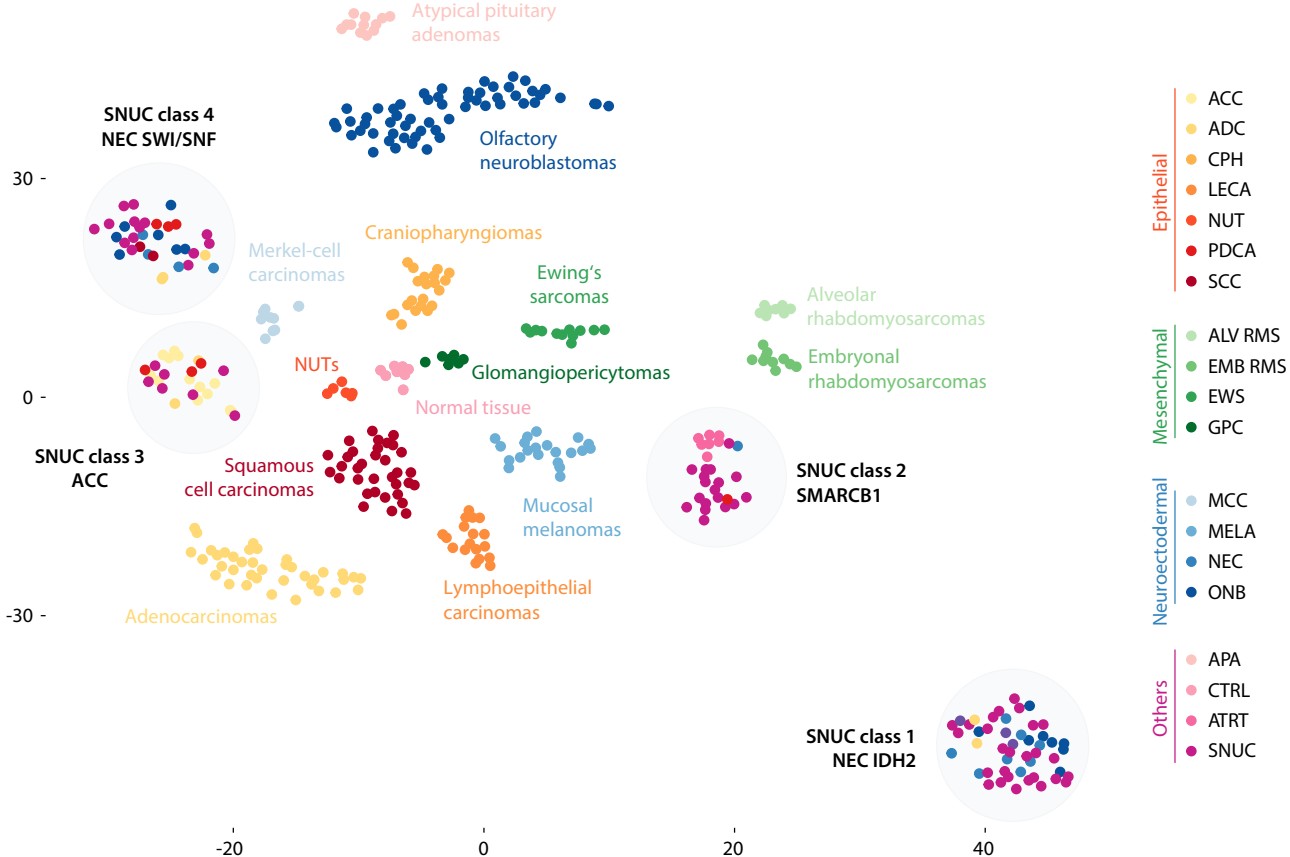

**Fig. 1 | DNA methylation classes of sinonasal tumors.** T-distributed stochastic neighbor embedding dimensionality reduction showing the 18 different DNA methylation classes. The conventional histopathological diagnosis is annotated by color. ACC sinonasal adenoid cystic carcinoma, ADC sinonasal adenocarcinoma, ALV RMS alveolar rhabdomyosarcoma, ATRT adult pituitary atypical rhabdoid/teratoid tumor, CPH craniopharyngioma, CTRL normal sinonasal control tissue,

EWS Ewing's sarcoma, EMB RMS embryonal rhabdomyosarcoma, GPC sinonasal glomangiopericytoma, LECA lymphoepithelial carcinoma, MCC Merkel-cell carcinoma, MELA sinonasal mucosal melanoma, NEC sinonasal neuroendocrine carcinoma, NUT NUT midline carcinoma, ONB olfactory neuroblastoma, PDCA sinonasal poorly differentiated carcinoma, PIT AD pituitary adenoma, SCC sinonasal squamous cell carcinoma, SNUC sinonasal undifferentiated carcinoma.

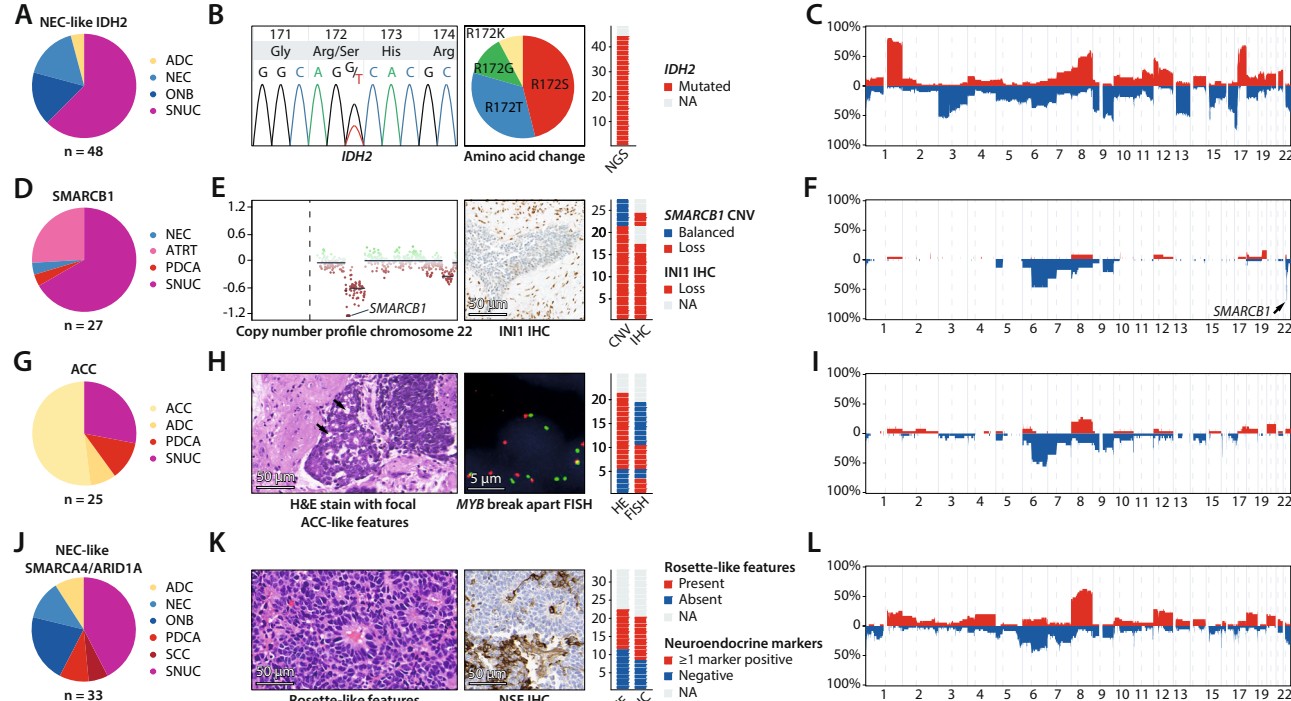

**Fig. 2 | Reclassification of sinonasal undifferentiated carcinoma (SNUC) classes.**
**A** Pie chart showing the conventional histopathological diagnosis of cases from the NEC-like IDH2 class. **B** Sanger-plot showing an example of an *IDH2* c.516 G > T (R172S) mutation. The frequencies of R172S, R172T, R172G and R172K mutations are displayed as a pie chart. A bar chart shows the frequency of R172 mutations, which occurred in all of the investigated cases. **C** Summary copy number profile of cases from the NEC-like IDH2 class showing highly recurrent copy number alterations such as gain of chromosome 1q as well as loss of chromosome 17p in combination with chromosome 17q gain. **D** Pie chart showing the conventional histopathological diagnosis of cases from the SMARCB1 class. **E** Detailed copy number plot of chromosome 22 with focal deletion of the *SMARCB1* gene locus and subsequent loss of INI1 expression in immunohistochemistry. **F** Summary copy number profile of cases from the SMARCB1 class, the *SMARCB1* gene locus is highlighted by the arrow. **G** Pie chart showing the conventional histopathological diagnosis of cases from the ACC class. **H** Example of histomorphological and molecular evidence for adenoid cystic (ACC) differentiation in form of sharply punched-out areas as well as recurrent *MYB* fusions. The frequency of these findings is shown as bar charts. **I** Summary copy number profile of cases from the ACC class. **J** Pie chart showing the conventional histopathological diagnosis of cases from NEC-like SMARCA4/ARID1A class. **K** Hematoxylin and eosin stain showing an example of rosette-like features that were recurrently observed in this class. The second tile shows an exemplary immunohistochemical stain for neuron-specific enolase (NSE) with very heterogenous cytoplasmic staining as example of evidence for neuroendocrine marker expression. The frequency of these findings is shown as bar charts. **L** Summary copy number profile of cases from the NEC-like SMARCA4/ARID1A class. ACC sinonasal adenoid cystic carcinoma, ADC sinonasal adenocarcinoma, ATRT adult sellar atypical teratoid/rhabdoid tumor, NEC sinonasal neuroendocrine carcinoma, ONB olfactory neuroblastoma, PDCA sinonasal poorly differentiated carcinoma, SCC sinonasal squamous cell carcinoma.

reviewed the available histopathological and molecular data on these cases.

The NEC-like IDH2 class (*n* = 48) contained tumors that had initially been diagnosed as either SNUCs, olfactory neuroblastomas, neuroendocrine carcinomas or adenocarcinomas (Fig. 2A). Molecular reports for these cases indicated a strong association with *IDH2* mutations. Additional mutational analysis confirmed that all cases with available tissue for testing harbored *IDH2* R172 hotspot mutations (Fig. 2B). Copy number profiles derived from DNA methylation data showed highly recurrent chromosomal aberrations, including gain of chromosome 1q as well as loss of chromosome 17p in combination with gain of chromosome 17q (Fig. 2C). Furthermore, tumors from this class showed a CpG island hypermethylation phenotype (Supplementary Fig. 6).

The SMARCB1 class (*n* = 27) consisted of histologically diagnosed SNUCs, neuroendocrine carcinomas, poorly differentiated carcinomas and atypical teratoid/rhabdoid tumors of adults in the sellar region (Fig. 2D)[21]. Tumors of this group were characterized by recurrent deletion of the *SMARCB1* gene locus (21/27; 80%) and subsequent loss of INI1 protein expression all cases with available tissue (16/16; 100%) including cases where the chromosomal *SMARCB1* loss was not identifiable (Fig. 2E). Apart from *SMARCB1* loss, we observed no additional highly recurrent chromosomal alterations (Fig. 2F). Based on these

findings, we conclude that inactivation of *SMARCB1* is the defining alteration for tumors from this DNA methylation subtype.

The ACC class (*n* = 25) mainly contained conventional adenoid cystic carcinomas (13/25; 52%), but also tumors that were initially diagnosed as adenocarcinomas, poorly differentiated carcinomas and SNUCs by means of conventional diagnostic criteria (Fig. 2G). By reevaluation of histomorphological areas, we found subtle adenoid cystic differentiation in two specimens that had initially been diagnosed as adenocarcinomas. Furthermore, FISH revealed *MYB* breaks prototypical for adenoid cystic carcinomas in three specimens with SNUC morphology (Fig. 2H). Based on these observations, we concluded that tumors from this class most likely represent histologically misclassified high-grade adenoid cystic carcinomas. Copy number profiling revealed few recurrent alterations, but loss of chromosome 6q was present in 56% (14/25) of samples (Fig. 2I).

The NEC-like SMARCA4/ARID1A class (*n* = 33) mainly consisted of tumors that had been diagnosed as SNUCs, neuroendocrine carcinomas and olfactory neuroblastomas, but also single adenocarcinomas, poorly differentiated carcinomas and squamous cell carcinomas (Fig. 2J). In our reevaluation, we observed rosette-like histological features in 50% of cases (11/22; Fig. 2K). Furthermore, 60% of these tumors (12/20) showed weak staining of at least one neuroendocrine marker (NSE, Chromogranin, Synaptophysin or CD56) in the initial

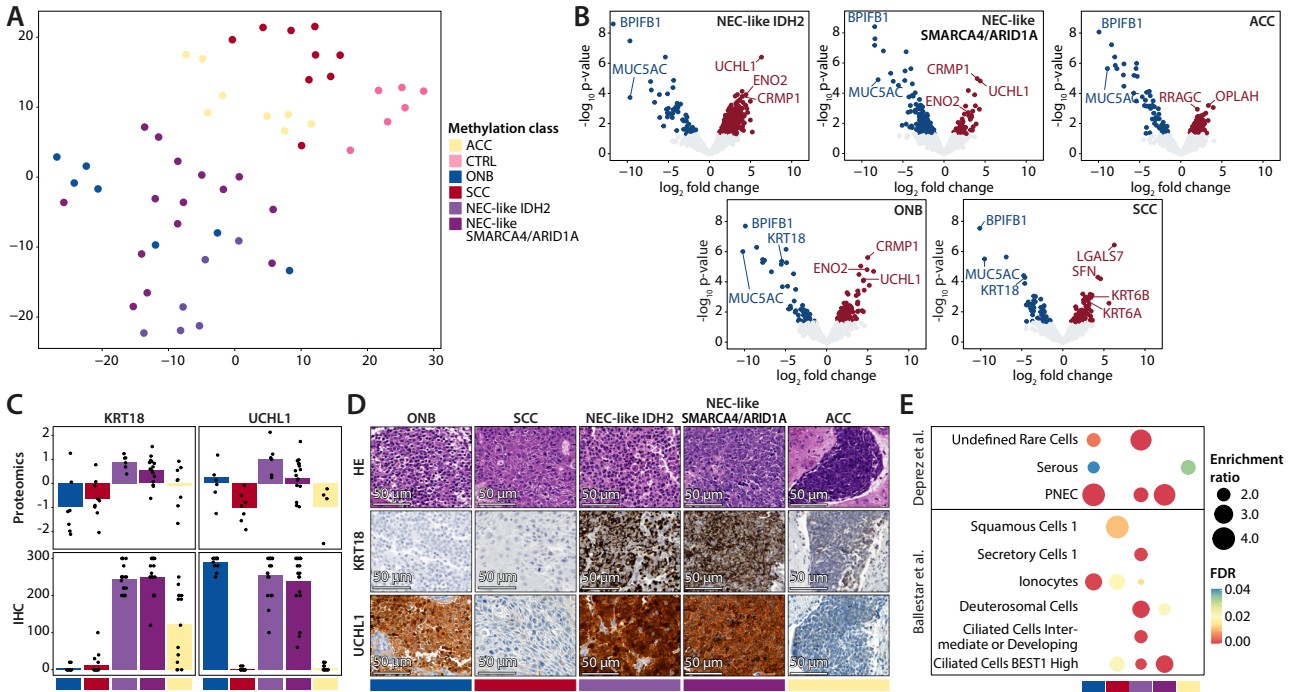

**Fig. 3 | Results from mass spectrometry-based proteomics. A** t-SNE depicting the global proteomic profile correlation between normal sinonasal tissue (CTRL), olfactory neuroblastoma (ONB), sinonasal squamous cell carcinoma (SCC), NEC-like IDH2, NEC-like SMARCA4/ARID1A, and adenoid cystic carcinoma (ACC). ONB, NEC-like IDH2 and NEC-like SMARCA4/ARID1A had an unexpectedly similar proteomic profile. **B** Differential expression analysis between normal sinonasal tissue and the investigated tumor classes was performed using a moderated t-test followed by Benjamini-Hochberg multiple testing correction. The results are shown as volcano plots. Recurrent and highly differential expressed proteins are annotated, highlighting overexpression of proteins specific for neurons or neuroendocrine cells in the ONB, NEC-like IDH2 and SMARCA4/ARID1A class. **C** Combined bar and point plots of 59 biologically independent samples showing the expression of Cytokeratin 18 (KRT18) and Ubiquitin carboxy-terminal hydrolase L1 (UCHL1) in the different tumor classes as determined by proteomics and immunohistochemistry (IHC). **D** Exemplary hematoxylin/eosin and KRT18 and UCHL1 immunohistochemical stainings are shown, validating the results from the proteomics analysis. In conclusion, the combination of these markers could be useful for

histopathological classification if DNA methylation is not available or feasible. **E** Results from overrepresentation analysis comparing the overall similarity of the global protein expression signatures of tumor classes with various normal cell types of the airways. Differentially expressed genes between normal sinonasal tissue and the investigated tumor classes were subjected to overrepresentation analysis using previously published cell type-specific gene sets that were identified using single-cell RNA sequencing data of the mucosal lining of human airways. The Fisher's exact test followed by FDR multiple testing correction was used to test for significance. A high similarity between pulmonary neuroendocrine cells ('PNEC') and the ONB, NEC-like IDH2 and NEC-like SMARCA4/ARID1A was observed, in line with the overexpression of neuronal/neuroendocrine markers shown in B. ACC specimens mostly resembled serous cells of submucosal glands and SCC specimens mostly resemble Squamous Cells 1. ACC adenoid cystic carcinoma, ADC adenocarcinoma, NEC neuroendocrine carcinoma, CTRL normal sinonasal tissue, ONB olfactory neuroblastoma, PDCA poorly differentiated carcinoma, SCC squamous cell carcinoma, FDR False discovery rate, PNEC Pulmonary neuroendocrine cells.

diagnostic workup. Summary copy number profiles revealed recurrent gain of chromosome 8q in up to 67% (22/33; Fig. 2L). Apart from their characteristic epigenetic profile, our initial review of the available sparse molecular data did not indicate recurrent or characteristic alterations. Further molecular analyses were thus performed.

**Mass spectrometry-based proteomics**
To identify characteristic protein expression profiles and potential cells of origins for specimens from the four SNUC classes, we performed mass spectrometry-based proteomics. Olfactory neuroblastomas, squamous cell carcinomas and normal sinonasal tissue were used as reference classes. Samples from the SMARCB1 class could not be included in this part of the study due to insufficient quantities of available tumor tissue.

T-SNE analysis of the most variably expressed proteins showed a pattern similar to that found by DNA methylation analysis (Fig. 3A). While normal tissue, squamous cell carcinomas and cases from the ACC tumor class were mostly assigned to distinct groups, the differentiation between olfactory neuroblastomas, cases from the NEC-like IDH2 and NEC-like SMARCA4/ARID1A class was less evident. As expected, differential expression analysis in comparison to normal sinonasal tissue revealed overexpression of classical neuronal proteins

in olfactory neuroblastomas (e.g., ENO2). NEC-like IDH2 and NEC-like SMARCA4/ARID1A tumors also demonstrated strong overexpression of proteins specific for neurons or cells of the diffuse neuroendocrine system such as UCHL1, CRMP1 and ENO2 (Fig. 3B), strongly indicating a neuroendocrine differentiation for both tumor classes. This pattern was not seen in tumors from the ACC class. In contrast, cytokeratin 18 (KRT18) was strongly overexpressed in both NEC-like IDH2 and NEC-like SMARCA4/ARID1A cancers but not in olfactory neuroblastoma. This predicts UCHL1 and KRT18 to be a potentially valuable marker combination for the differentiation of olfactory neuroblastomas and the SNUC classes. We performed an immunohistochemical validation of this marker combination (Fig. 3C) and observed strong staining of KRT18 in all cases of the NEC-like IDH2 and NEC-like SMARCA4/ARID1A class, variable staining intensity in the ACC class and no staining in all investigated olfactory neuroblastomas and most squamous cell carcinomas (Fig. 3D). UCHL1 expression was high in NEC-like IDH2 and NEC-like SMARCA4/ARID1A tumors as well as olfactory neuroblastomas but absent in tumors from the ACC class and squamous cell carcinomas. We thus concluded that the combination of both markers could be of diagnostic value for tumor classification.

To identify potential cells of origin, differentially expressed proteins from all tumor classes in comparison to normal sinonasal tissue

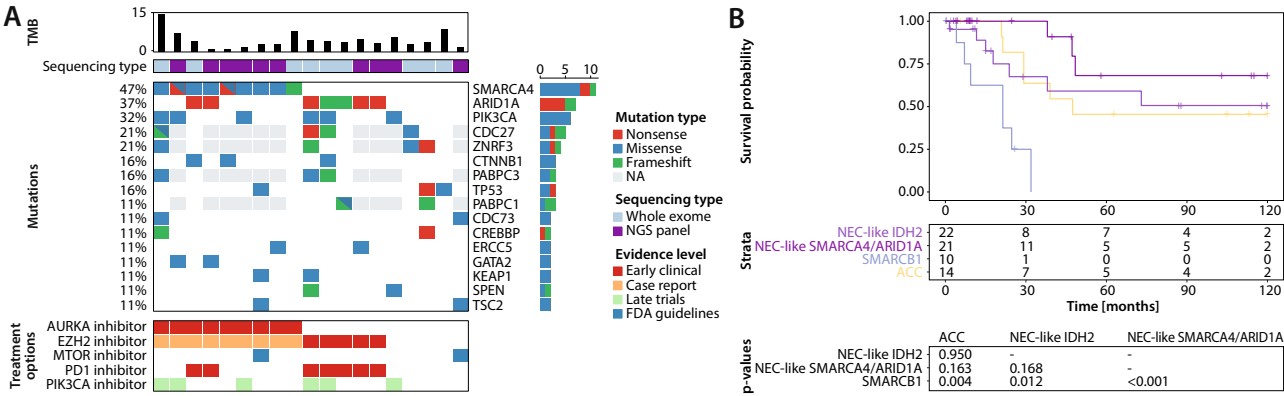

**Fig. 4 | DNA sequencing results for NEC-like SMARCA4/ARID1A tumors and outcome analysis of sinonasal undifferentiated carcinoma (SNUC) DNA methylation classes. A** The oncoprint plot shows tumor mutation burden (TMB) and recurrent mutations from whole exome sequencing or next generation panel sequencing (NGS). Potential treatment options and their respective evidence levels according to the classification from the Cancer Genome Interpreter website are shown in the panel below. **B** Overall survival Kaplan–Meier curve comparing disease-specific survival rates in different SNUC DNA methylation classes. The tables below show the number of patients at risk at different time points as well as a pairwise test for significance using the log-rank test. Patients with tumors from the SMARCB1 class had significantly worse survival compared to adenoid cystic carcinoma (ACC; $p = 0.004$), NEC-like IDH2 ($p = 0.012$) and NEC-like SMARCA4/ARID1A ($p < 0.001$) tumors. There was no significant difference between tumors from the NEC-like SMARCA4/ARID1A and ACC ($p = 0.163$) and NEC-like IDH2 ($p = 0.168$) class as well as between ACC and NEC-like IDH2 tumors ($p = 0.950$). NEC neuroendocrine carcinoma.

were subjected to overrepresentation analysis using cell type-specific gene sets from previously published single cell RNA sequencing data of the mucosal lining of upper and lower human airways (Fig. 3E)[22,23]. The neuroectodermal differentiation of specimens from the NEC-like IDH2 (FDR < 0.001) and NEC-like SMARCA4/ARID1A class (FDR < 0.001) as well as olfactory neuroblastomas (FDR < 0.001) was reflected in their similarity with pulmonary neuroendocrine cells (PNEC). Furthermore, olfactory neuroblastomas (FDR 0.004) and tumors from the NEC-like IDH2 group (FDR < 0.001) showed similarity to a class of Undefined Rare Cells, which likely represent progenitor cells of epithelial and neuroendocrine cells[22]. Tumors from the ACC class mostly resembled serous cells of submucosal glands (Serous; FDR 0.029). As expected, squamous cell carcinoma profiles were closely related to squamous cells (Squamous Cell 1; FDR 0.011).

Additionally, we performed a differential protein expression analysis comparing the ACC, NEC-like IDH2 and NEC-like SMARCA4/ARID1A tumor classes against each other. Protein lists were subjected to functional pathway analysis (Supplementary Fig. 7). Tumors from the NEC-like IDH2 class were enriched for several functional terms related to mitochondrial processes, including proteins related to the citric acid cycle. ACC class tumors showed evidence for alterations in MAPK-related signaling pathways while the few significant functional terms for cases from the NEC-like SMARCA4/ARID1A class were mainly associated with translational processes.

### Mutational profiling of the NEC-like SMARCA4/ARID1A methylation class

As a clear driver for the NEC-like SMARCA4/ARID1A molecular tumor class was not apparent in the available retrospective data, we performed whole exome ($n = 9$) or NGS panel sequencing ($n = 10$) of 19 tumors from this group. We observed relatively low median tumor mutational burden with 3.7 mutations per megabase (Fig. 4A). High mutational rates were seen in genes involved in the formation of the SWI/SNF chromatin remodeling complex (14/19; 74%), including *SMARCA4* (9/19; 47%) and *ARID1A* (7/19; 37%). Early clinical data suggests that patients with these alterations might benefit from treatment with PD1 inhibitors[24]. Notably, we observed one case of a young patient in which a *SMARCA4* frameshift mutation (p.Q306Rfs*12) was detected in tumor and in adjacent normal tissue, suggesting a germline or mosaic origin for this mutation. Additional recurrent alterations in this tumor class comprised *PIK3CA* mutations (6/19; 32%), including

classical hotspot mutations such as p.H1047 or p.E545, which are known predictive markers for treatment with PIK3 pathway inhibitors in breast cancer[25,26]. Other known pathogenic driver mutations included *CTNNB1* (3/19; 16%), *TP53* (3/19; 16%) and *TSC2* (2/19; 11%).

### Clinical implications of DNA methylation classes

To further evaluate the clinical importance of the DNA methylation-based classes, we compared disease-specific survival between the four SNUC classes (Fig. 4B). SMARCB1 class tumors were associated with significantly worse disease-specific survival compared to cases from the NEC-like IDH2 ($p = 0.012$), NEC-like SMARCA4/ARID1A ($p < 0.001$) or ACC class ($p = 0.004$). Best survival rates were seen in the NEC-like SMARCA4/ARID1A group, although there was no significant difference in comparison with the other two classes (vs. ACC: $p = 0.163$; vs. NEC-like IDH2: $p = 0.168$). In the ACC methylation class, we observed no significant difference in survival between tumors that were classified as adenoid cystic carcinoma or as SNUC by conventional histopathology ($p = 0.5$).

### Machine learning classifier development

As described above, we used t-SNE and hierarchical clustering to define epigenetic classes. However, t-SNE is not a reliable tool to classify new cases as the position of individual data points can change over different iterations and highly depends on selected parameters and the composition of the cohort. For convenient and rapid classification of raw DNA methylation data in a diagnostic setting, we used the data from the reference set to develop a machine learning algorithm that assigns a given sample to one of the DNA methylation classes. We also implemented a supervised outlier detection designed to recognize and prevent the classification of samples with divergent DNA methylation profiles, such as distant metastases from other organs or entities that have not been included in the development of the classifier. For this, we collected a set of 8065 tumor and normal samples, covering eight different categories (e.g., adenocarcinoma) including 197 different exact diagnoses (e.g., colorectal adenocarcinoma). A subset of 400 cases from this cohort was used for the training of an additional Unknown class. The performance of the classifiers was validated using an independent test set consisting of 52 sinonasal tumors as well as the remaining 7665 non-sinonasal tumors.

To explore the most suitable technique for this classification task, we compared support vector machine[27–29] and random forest[30]

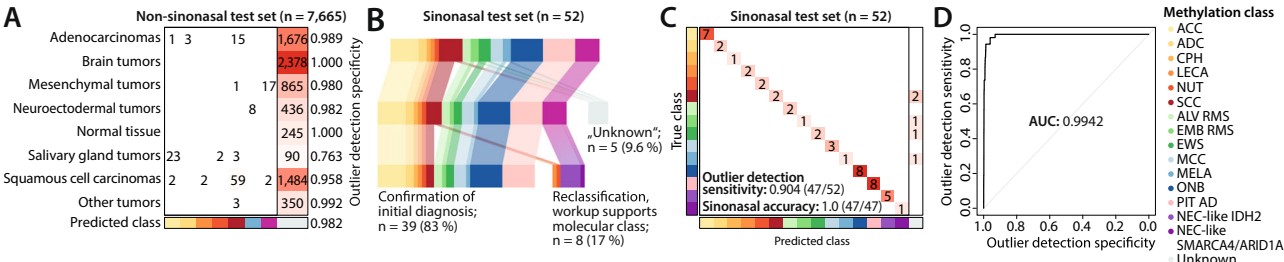

**Fig. 5 | Classifier development and application to an independent test set.**
**A** Confusion matrix of the non-sinonasal samples from the test set, split up in different categories. Overall, the classifier achieved a specificity of 0.982, ranging from 0.763 in salivary gland tumors to 1.0 in normal tissue and brain tumors.
**B** Overview of the classification results from the test set. Out of 52 samples, five cases were assigned to the Unknown class, resulting in an outlier detection sensitivity of 0.904. Out of the 47 remaining specimens, DNA methylation-based classification confirmed the initial diagnosis in 39 samples (83%). Eight specimens (17%) were reclassified to a divergent DNA methylation class. Additional molecular workup supported the DNA methylation-based reclassification in all cases.
**C** Confusion matrix from the sinonasal test set shows an accuracy of 1.0 for classification of sinonasal tumors. **D** Receiver operating characteristic (ROC) curve of the support vector machine classifier with regards to the binary outlier detection problem with an area under the curve (AUC) of 0.9942.

machine learning algorithms, the latter being the current gold standard for DNA methylation-based tumor classification[14,16,31]. In a hypothetical diagnostic setting, the potential hazard of a false classification is higher than the hazard of an unsuccessful classification. While the random forest achieved higher sensitivity values, the results of the support vector machine were superior with regards to specificity and accuracy. Therefore, the support vector machine will be further described in detail. For comparison, the results of the random forest classification are shown in Supplementary Table 1.

We evaluated the performance of the classifier using three different metrics. The algorithm demonstrated a high specificity of 0.982 (7524/7665) to correctly assign non-sinonasal tumor specimens to the Unknown class. We observed some variation in the specificity in different categories of the non-sinonasal test set (Fig. 5A). The lowest values were observed in salivary gland tumors (0.763) and the highest values in brain tumors (1.0) as well as normal tissue (1.0). Of note, 107 of the 197 exact non-sinonasal diagnoses (55.2%) were exclusively present in the test and not in the reference set. The classifier achieved only slightly higher specificities for diagnoses that were included in both sets (6,402/6,492; 0.986) compared to diagnoses that were exclusive to the test set (11,22/1,173; 0.957), demonstrating its reliability to recognize unseen data types. The overall sensitivity to identify primary sinonasal tumors was 0.904 (47/52; Fig. 5B). In 39 of the 47 sinonasal tumor specimens (83%), the DNA methylation-based classification confirmed the initial histopathological diagnosis. Two SCCs were assigned to the LECA and the NUT DNA methylation class and the molecular classification was confirmed by positive EBV-encoded RNA (EBER) in-situ hybridization and positive RNA-based *NUTM1* fusion analysis, respectively. Furthermore, the sinonasal tumor set also contained six SNUC specimens. Five of these were classified as NEC-like IDH2 and subsequent mutational analysis revealed the presence of an *IDH2* R172 mutation in all cases. The remaining SNUC specimen was assigned to the NEC-like SMARCA4/ARID1A class and DNA sequencing confirmed a truncating *SMARCA4* mutation (p.Q611*). Furthermore, all six samples showed strong expression of UCHL1. Thus, the molecular workup confirmed the DNA methylation-based diagnosis in all reclassified cases. The classifier, therefore, achieved an accuracy of 1.0 on the sinonasal validation cohort and lead to a revision or refinement of the initial diagnosis in 17% of cases (Fig. 5C, D).

A web platform which provides convenient access to the classification algorithm can be accessed at www.aimethylation.com.

## Discussion

In this study, we provide a resource of DNA methylation profiles from a diverse cohort of sinonasal tumors and present a machine learning algorithm for a robust classification of these diagnostically challenging tumors. Using DNA methylation profiling, DNA sequencing, copy

number analysis and mass spectrometry-based proteomics, we show that tumors with SNUC morphology are not as undifferentiated as their current terminology suggests, but rather consist of four different molecularly distinct entities.

A cohort of the clinically relevant spectrum of sinonasal tumors and associated neoplasms was surveyed for DNA methylation classification. In line with previous studies from other fields, we were able to demonstrate that most established tumor entities show characteristic DNA methylation signatures which can be used for reliable clinical classification and differentiation[14–18,32]. While earlier studies only covered a fraction of the sinonasal cancer spectrum with rather limited total case numbers, the current study includes the whole spectrum of diagnostically relevant tumor classes and has clearly increased the total numbers of samples. This allowed to identify previously unrecognized DNA methylation-based tumor classes among sinonasal tumors. During the assembly of the reference cohort, 34 specimens were excluded, as they could not be stably assigned to an epigenetic tumor class. There are several aspects that could explain this. First, some of the excluded cases encompassed diagnoses with insufficient number of cases to form a separate and stable class (e.g., biphenotypic sinonasal sarcoma). These entities could be included in future versions of the classifier, if additional cases can be acquired. Second, slightly divergent DNA methylation profiles could also be caused by array quality or by technical variations between different analyses. Third, some of these cases could also be of non-sinonasal origin, such as advanced tumors from neighboring anatomic regions (e.g., tumors originally arising from the palate or brain) with continuous infiltration of sinonasal structures or distant metastases from an unrecognized primary site. Of note, none of these tumors were used for the development of the classifier as they did not correspond to a stable epigenetic class and were therefore excluded from further analyses. Finally, other cases of these non-clustering samples could correspond to hitherto unrecognized, even rarer tumor classes that require additional investigation. The last two points could also explain the large proportion of neuroendocrine carcinomas and SNUCs in the noise point category. Non-sinonasal tumors would be prone to be histologically classified as SNUCs due to their unusual morphology. Furthermore, expression of neuroendocrine markers is not uncommon in advanced and potentially dedifferentiated carcinomas, making the classification as neuroendocrine carcinoma more likely. To facilitate the further characterization of potentially unrecognized classes, we provide the unprocessed DNA methylation data for these cases along with the data of the reference cohort.

Using an independent test set, we were able to show that the DNA methylation-based classification algorithm can reliably subtype samples with SNUC morphology without the need for additional molecular testing. Furthermore, the classifier correctly reclassified two samples

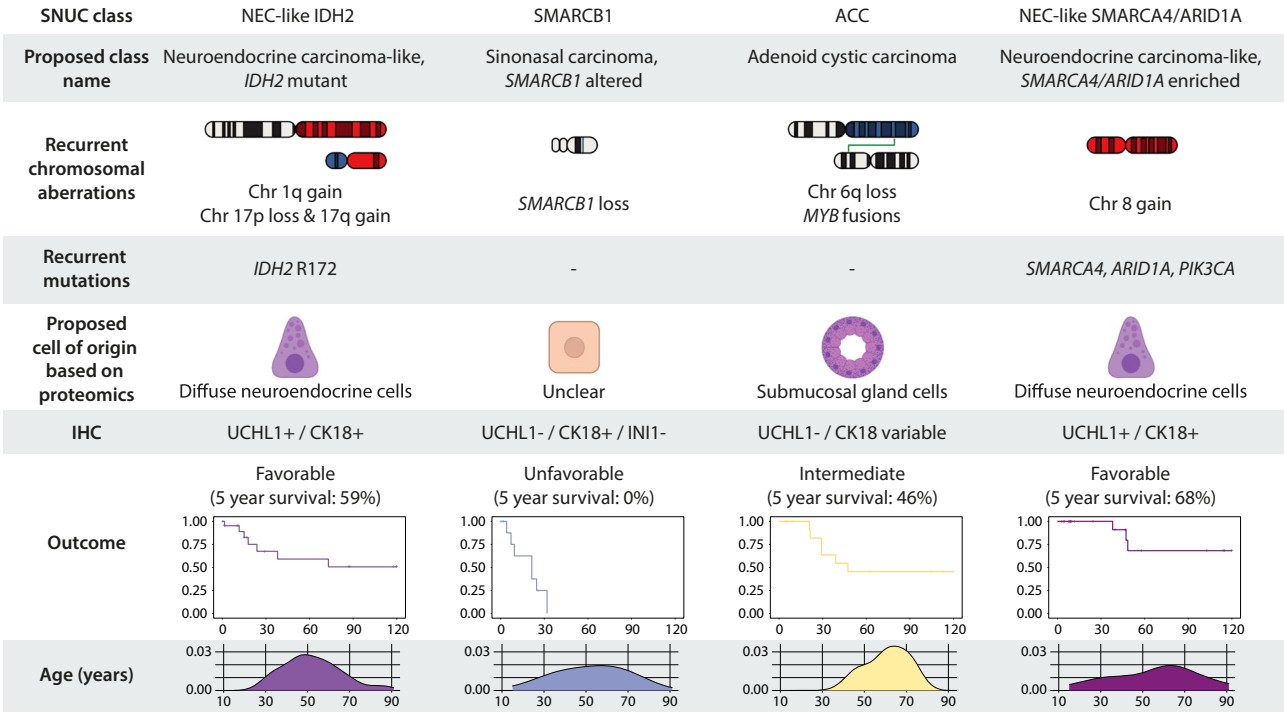

**Fig. 6 | Summary of important molecular characteristics and clinical parameters of the four sinonasal undifferentiated carcinoma (SNUC) classes, including proposed type names and potential immunohistochemical (IHC)** **markers.** SNUC Sinonasal undifferentiated carcinoma, Chr Chromosome, UCHL1 Ubiquitin carboxy-terminal hydrolase L1, CK18 Cytokeratin 18.

initially diagnosed as sinonasal squamous cell carcinomas as lymphoepithelial carcinoma and NUT midline carcinoma, respectively. This reclassification is of profound clinical importance, as lymphoepithelial carcinomas show improved response to radiotherapy while NUT midline carcinomas are associated with very poor prognosis. We also describe the implementation of a supervised outlier detection for enhancing DNA methylation-based tumor classification. Machine learning algorithms typically assign anomalous profiles to the next, most similar class, potentially leading to spurious classification results. In the context of DNA methylation-based classification, entities that have not been used for the training of the algorithm such as distant metastases from other organs would thus either go unnoticed or be assigned to a wrong class. Crucially, both errors can be avoided by incorporating outlier detection in the machine learning pipeline. Although this approach may slightly compromise sensitivity, it reduces the risk of misclassifying non-sinonasal tumors, which is crucial for application in a potential diagnostic setting. Copy number profiles derived from DNA methylation data revealed different recurrent copy number alterations between the tumor classes. This information may provide additional confidence to the DNA methylation-based classification, although the sensitivity and specify of these alterations for classificatory purposes seem limited.

A further focus of our study was to investigate different DNA methylation signatures and molecular alterations in the diagnostically highly challenging group of SNUCs. Our results indicate that what is currently summarized as SNUCs likely represent a heterogeneous group of tumors comprising at least four different molecular classes with different molecular drivers and different clinical course. A summary of the most important characteristics of the four classes is provided in Fig. 6.

Specimens from the NEC-like IDH2 class were characterized by *IDH2* mutations and highly recurrent copy number alterations. Similar to acute myeloid leukemia and gliomas, *IDH2* mutations induce a CpG island hypermethylation phenotype in sinonasal carcinomas, resulting

in a highly distinct DNA methylation signature and significant hypermethylation of various tumor-related genes[33]. The value of *IDH2*-specific inhibitors in the treatment of patients with sinonasal tumors is currently unknown[34]. In line with previous reports, patients with sinonasal tumors with *IDH2* mutations have a comparably favorable prognosis[35].

Most cases from the NEC-like SMARCA4/ARID1A class showed *SMARCA4* or *ARID1A* mutations, which are part of the SWI/SNF chromatin remodeling complex. This also included one case with a *SMARCA4* loss of function mutation in tumor-free normal tissue, potentially representing a germline or mosaic mutation. Germline mutations of *SMARCA4* have previously been described to be associated with Rhabdoid tumor predisposition syndrome 2, leading to highly aggressive and early-onset tumors such as small cell carcinoma of the ovary, hypercalcemic type[36]. Although our data is clearly limited in this aspect, our findings suggest that NEC-like SMARCA4/ARID1A tumors may also occur in the context of tumor predisposition syndromes. Furthermore, we also observed a remarkably high rate of activating and potentially actionable *PIK3CA* mutations in almost one-third of NEC-like SMARCA4/ARID1A tumors. Comparably high mutation rates of *PIK3CA* are observed in breast carcinomas but have so far not been detected in other types of cancer[26]. There have been previous studies which observed activating *PIK3CA* mutations in SNUCs at low frequency, which indicates that these alterations are likely enriched in NEC-like SMARCA4/ARID1A tumors[8,11]. Functional studies or clinical trials will be required to evaluate whether these tumors may be responsive to treatment with PIK3 pathway inhibitors. Overall, the prognosis of patients from this tumor class is relatively favorable and comparable to *IDH2* mutated tumors.

In mass spectrometry-based proteomics, we observed relatively similar global protein expression profiles for NEC-like IDH2 and NEC-like SMARCA4/ARID1A tumors. In both classes, we identified over-expression of several proteins that are specific to neurons or cells of the diffuse neuroendocrine system, and gene set enrichment analysis

also indicated a high similarity with neuroendocrine cells. The identified markers are not routinely established in histopathological laboratories, ENO2 being a possible exception, and have therefore not been investigated in previous studies or used in routine diagnostics. Importantly, routinely used diagnostic markers such as chromogranin A or synaptophysin were not among the highly enriched markers in our proteomics analysis and may thus fail to identify the neuroendocrine differentiation of these cancers. This might explain why a substantial proportion of tumors from this class were diagnosed as SNUCs or even as adenocarcinomas or squamous cell carcinomas. It should further be mentioned that in our extensive reference cohort of sinonasal tumors no other class of neuroendocrine carcinomas could be identified. Based on our findings, we therefore, propose that these tumors should be regarded as "neuroendocrine carcinoma related", either characterized by *IDH2* mutations (neuroendocrine carcinoma-like, *IDH2* mutant) or recurrent *SMARCA4/ARID1A* alterations (neuroendocrine carcinoma-like, *SMARCA4/ARID1A* enriched). However, it must be noted that this concept may change the treatment of patients with tumors of SNUC morphology. Therefore, a careful clinical evaluation and confirmation in further studies is crucial before drawing any clinical conclusions from our study.

With regards to routine histopathological workup, we identified KRT18 in combination with UCHL1 as potential immunohistochemical markers to differentiate NEC-like IDH2 and NEC-like SMARCA4/ARID1A tumors from adenoid cystic carcinomas, olfactory neuroblastomas and squamous cell carcinomas. The combinational use of these markers could be of high diagnostic value when DNA methylation analysis is not available or not feasible.

For tumors from the NEC-like IDH2 class, functional analysis of proteomic data revealed alterations in mitochondrial processes, including the citric acid cycle. This is in line with the well-known oncogenic mechanism of mutated *IDH1/2*, disrupting the citric acid cycle which is located in the inner mitochondrial membrane and producing the oncometabolite 2-hydroxyglutarate[37]. For NEC-like SMARCA4/ARID1A tumors, we observed a general association with translational processes, however, no specifically disrupted pathways were observed.

Cases from the SMARCB1 class were characterized by *SMARCB1* deficiency, which has recently been identified among SNUCs[10,19]. Our data further substantiates that sinonasal tumors with this alteration represent a distinct entity, including a broad range of histological morphologies and should therefore be identified by molecular testing. Although we were not able to include tumors from this class in our proteomics study due to insufficient quantities of tumor tissue, we did not detect immunohistochemical expression of the neuroendocrine and neuronal markers that were upregulated in NEC-like IDH2 and NEC-like SMARCA4/ARID1A tumors. This suggests a different cell of origin for these cancers and further studies are required to clarify their origin. Interestingly, *SMARCB1*-deficient sinonasal carcinomas show a remarkable epigenetic similarity to adult sellar atypical teratoid/rhabdoid tumors, although they tended to aggregate slightly separate in t-SNE analysis. It remains unclear if this is due to batch effects or if these tumors actually represent two distinct tumor types sharing the same driving alteration.

Samples from the ACC class shared the molecular profile (DNA methylation, MYB rearrangement, recurrent loss of chromosome 6q[38]) of adenoid cystic carcinomas and most likely represent high-grade adenoid cystic carcinomas. In several such tumors, we also detected focal adenoid cystic differentiation on histological reexamination. In addition, mass spectrometry-based proteomics revealed similarities of these tumors with serous cells of submucosal glands, further supporting the reclassification. Functional analysis of proteomic data revealed evidence of MAPK-pathway activation as a key mechanism which is in line with other reports[39,40]. Previous studies recognized that solid variants of adenoid cystic carcinomas can be mistaken for SNUCs

and that close histomorphological investigation and adequate sampling is crucial[41]. A major benefit of DNA methylation-based classification is that it does not require the analysis of a tumor area with a certain differentiation or growth pattern. Therefore, a classification is also possible if only high-grade tumor areas are available (e.g. in smaller biopsy specimens or partial resections).

The findings of our study come with some limitations. While numerous studies showed that DNA methylation is a very reliable tool for tumor classification, the underlying biological mechanisms remain relatively unclear. In our study, the CpGs relevant for classification showed a very similar distribution over chromosomes and functional gene regions compared to the overall array design. Furthermore, most relevant CpGs were located in the gene body. The regulatory effect of DNA methylation in these regions is only poorly understood and interpretation is not straightforward.

Second, the main goal of our proteomic analysis was to identify potential diagnostic markers and cells of origin. While the selected LFQ approach was suitable to accomplish these tasks, mechanistic analyses focusing on less abundant signaling pathway molecules would profit from more sensitive approaches such as data-independent acquisition (DIA) or tandem mass tag (TMT) labeling as well phosphoproteomic profiling. Therefore, the results from our functional analyses should be interpreted with caution and should be further investigated in future studies.

Third, we did not perform central histopathological review of the cases included in this study. Therefore, the quality of the given conventional diagnoses might differ between the providing institutions due to different expertise in the diagnosis of sinonasal tumors.

Furthermore, our outcome analysis should be interpreted with caution, as there was only very limited data available on other outcome associated clinical factors such as local tumor stage or metastatic stage.

In summary, we provide a DNA methylation-based algorithm, which could serve as a valuable tool in the diagnosis of sinonasal tumors, preventing misclassifications and supporting the workup of challenging cases. In addition, we clarify the molecular heterogeneity of tumors with SNUC morphology. We demonstrate that tumors with SNUC morphology can be segregated to four distinct tumor types, including (1) sinonasal neuroendocrine carcinoma-like, *IDH2* mutant, (2) sinonasal neuroendocrine carcinoma-like, *SMARCA4/ARID1A* enriched, (3) sinonasal carcinoma, *SMARCB1* altered and (4) poorly differentiated adenoid cystic carcinoma.

## Methods

### Ethics statement
This research project has been approved by the ethics committee of the Charité – Universitätsmedizin Berlin. Retrospective investigation of left-over diagnostic samples for research purposes was covered by the general treatment agreement of the respective hospitals. No compensations were provided.

### Statistics & Reproducibility
Statistical analysis was performed in RStudio Version 1.3.1093. For the reference cohort, minimum sample size per histopathological entity was set at 6, similarly to previously published work[14]. Detailed parameters for exclusion of low-quality samples and cases that did not correspond to a stable DNA methylation class are listed in the DNA methylation analysis section. The investigators were not blinded to allocation during experiments and outcome assessment, but the algorithms that assigned cases to DNA methylation classes were agnostic to the conventional histopathological diagnosis.

### Sinonasal reference cohort
For the identification of DNA methylation-based tumor classes and the development of corresponding machine learning classifiers, we

compiled a reference cohort of 495 samples from sinonasal tumors (Supplementary Data 2). 271 formalin-fixed and paraffin embedded (FFPE) tissue specimens were retrieved from the archives of the Institutes of Pathology or Neuropathology at the University Hospitals Basel, Berlin, Frankfurt am Main, Gießen, Göttingen, Hamburg, Heidelberg, Marburg, München, Münster, Naples, Oviedo, Lübeck, Stanford and Tübingen. The conventional histopathological diagnosis was taken from the original histology report of the providing center or the associated metadata if the samples was derived from a previously published study. Specimens were not reviewed centrally prior to inclusion. Normal sinonasal tissue samples were retrieved from independent patients undergoing sinonasal surgery due to non-neoplastic conditions. All samples were histologically evaluated and confirmed to be free of tumor before DNA extraction. Raw IDAT files from an additional 190 samples from previously published studies were retrieved from public repositories or provided by the authors[10,14,16,19,21]. 32 samples were excluded after quality control, including 20 samples with poor DNA methylation analysis quality metrics as well as 12 specimens with low tumor cell content. 429 cases were used for subsequent analyses. Access to FFPE blocks to reproduce or validate the findings described in this manuscript can be obtained if sufficient material is left for further analyses.

### Test set

An additional, independent cohort of 52 sinonasal tumors was compiled as a test set for the validation of the machine learning classifiers (Supplementary Data 3). The samples in this cohort were neither used in the identification of methylation classes nor in the development of the classifiers, nor for dimensionality reduction.

### Cohort of non-sinonasal tumors

For the implementation of an outlier detection, we compiled a cohort of 8104 tumor and normal tissue samples covering 197 different diagnoses which we further grouped into eight categories. Raw DNA methylation data in form of IDAT files were retrieved from publicly available repositories as well as our own analyses from other research projects[14,16]. In a quality control, 39 samples were excluded from further analysis. The final cohort was randomly split in two cohorts and its samples either used for the development or the evaluation of the classifiers. All samples included in the non-sinonasal tumor cohort are listed in Supplementary Data 4.

### Immunohistochemistry

Immunohistochemical staining was performed on the BenchMark XT (Ventana) automated slide stainer according to the manufacturer's instructions. Sections were incubated with primary antibody against UCHL1 (clone 13C4, dilution 1:1000, abcam, United Kingdom, catalog number ab8189) and KRT18 (clone DC-10, dilution 1:1000, BioGenex, USA, catalog number AM143-5M). Antibodies were validated using adequate positive controls, including human neural tissue for UCHL1 and human cancer tissue for KRT18. Expression was scored using an H-score which was calculated by multiplying the staining intensity (0: no staining; 1: weak staining; 2: moderate staining; 3: strong staining) by the respective percentage of tumor cells[42].

### In-situ hybridization

Fluorescence in-situ hybridization (FISH) was performed as described previously[43] using the MYB Dual Color Break Apart Probe (Zytovision). In brief, 4 μm sections were deparaffinized dehydrated and incubated in pretreatment solution (Dako, Denmark) at 95–99 °C for 10 min. Following immersion in pepsin solution for 3–6 min at 37 °C, slides were washed, dehydrated and air dried. DNA probes were applied and the sections were sealed and denaturalized in humidified atmosphere at 82 °C for 5 min. Sections hybridized at 45 °C overnight. After washing, slides were counterstained with 4′,6-diamidino-2-phenylindole (DAPI).

Silver-enhanced in-situ hybridization for EBV analysis was done using the BOND Epstein-Barr virus-encoded small RNA (EBER) Probe (Leica) on the BOND-MAX automated slide stainer (Leica).

### DNA extraction

Representative tumor areas were identified using light microscopy of hematoxylin and eosin-stained sections. Semi-automated DNA extraction was performed on the Maxwell RSC Instrument using the Maxwell RSC FFPE Plus DNA Purification Kit (Custom, AX4920; Promega). Extracted total DNA quantities were measured using the Qubit™ HS DNA Assay (Thermo Fisher Scientific).

### DNA methylation analysis

We used the Illumina Infinium HD FFPE DNA Restore Kit for DNA restoration of FFPE samples. Subsequent bisulfite conversion was performed using the EpiTect Bisulfite Kit (Qiagen). The bisulfite-converted DNA was analyzed using the Illumina Infinium HumanMethylation450 or MethylationEPIC BeadChip.

Raw DNA methylation data were processed in RStudio Version 1.3.1093 using the minfi package[44]. The *pfilter* (with perc = 5) and *rmSNPandCH* functions from the wateRmelon and DMRcate packages were used to exclude low-quality samples and to filter CpGs with low quality, reported cross-reactivity or association with SNPs or sex chromosomes[45,46]. The 20,000 most variant CpG sites were selected for further analysis. The *combineArrays* function of the minfi package was used to merge EPIC and 450k data. T-SNE was done using the RTSNE package, using a perplexity of 20 and 4000 iterations[47]. Density-based spatial clustering of applications with noise (DBSCAN) with the minPts parameter set at 6 was used to determine the optimal number of classes based on t-SNE coordinates and to assign individual cases to their respective class. Cases that were labeled as noise points were excluded from further analysis. Comparison of the number of classes and the assignment of the non-outlier cases to these classes revealed no differences before and after exclusion of the outlier samples. Robustness of tumor classes derived from t-SNE analysis was tested using iterative random down-sampling to 80% of the total cohort, as described previously[14]. The Pearson's correlation coefficient of the x and y coordinates for all samples were calculated after 300 iterations. Tumor purity was estimated using the *predict_purity_betas* function[48].

### Classifier development

We developed two separate machine learning classifiers based on a support vector machine and a random forest model that predict the tumor class of sinonasal tumor samples from their DNA methylation profile. In addition to these classes, a single non-sinonasal class for other tumor entities was introduced to detect outliers. Outlier detection has several modes of operation, namely unsupervised (where no outlier labels are required), semi-supervised (where only a few outlier labels are available) and supervised outlier classification. The latter was used in this study to distinguish sinonasal tumors from all other non-sinonasal tumors[49].

The models were developed on a training set composed of all samples from the sinonasal reference cohort (n = 395) and 5% of the samples of each category from the non-sinonasal cohort (n = 400), which were randomly selected. This resulted in a combined dataset of 795 samples.

On this combined training set, the optimal hyperparameters for both model types were then determined in a grid search by minimizing the class-balanced multinomial cross-entropy loss in a five-fold cross-validation with stratified sampling. A dimension reduction to the 20,000 most variant CpG sites was performed on each training set of the cross-validation and applied to the respective validation fold. The

final models were then retrained on the full training set with the selected hyperparameters.

For the development of support vector machine models, we used the R package e1071. Linear and radial basis function kernels, gamma values of $\gamma = 2^{-3,...,3} / 20,000$, and cost parameters of $C = 2^{0,...,5}$ were considered as possible hyperparameters. Random forest models were trained with the R package randomForest, using the number of trees ntree=500, 1000 and mtry=$2^{-5,...,5}$ x sqrt(20,000) as hyperparameters. Further, both models were configured to return scores for each methylation class.

In order to make these scores more readily interpretable as probabilities, we developed calibration models based on ridge-penalized multinomial logistic regression, resembling previously described procedures but accounting for the challenge of a lower number of samples here[31]. In detail, we used the *cv.glmnet* function of the R package glmnet on the scores on the training set resulting from the previous cross-validation which correspond to the selected hyperparameters. For prediction of the calibrated scores, the λ parameter with minimum mean cross-validated error was chosen. The class with the highest calibrated score was then determined as the final prediction for each sample.

The resulting classification procedure was then evaluated on the sinonasal test cohort ($n = 52$) and the samples from the non-sinonasal cohort that had not been included in the training set before ($n = 7,665$). In order to assess the outlier detection, all predictions of sinonasal tumor classes were retrospectively combined in one class and sensitivity and specificity were computed for the binary differentiation of sinonasal and non-sinonasal samples (outlier detection specificity and outlier detection sensitivity). For the evaluation of the sinonasal methylation class prediction, only samples from the sinonasal test cohort without classification as Unknown were considered (sinonasal accuracy). Five repetitions of the classifier development and evaluation led to similar results as with our final classifier, confirming the stability of the procedure.

## Copy number analysis

Genome-wide copy number profiles were generated from raw DNA methylation data using a modified version of the *conumee* package[50,51].

## Mass spectrometry-based proteomics

Sufficient tissue for mass spectrometry-based proteomics was available for 66 cases, including 59 tumor samples and seven normal sinonasal tissue specimens.

Representative 1.0 or 1.5 mm punch biopsy needle tissue cores were subjected to sonication using a Covaris LE220Rsc Focused-ultrasonicator (250 W, 50% duty cycle, 3 rounds with incubation at 80 °C for 1 h/ 95 °C for 30 min between the rounds) in a denaturing buffer containing 1.5% SDS and 2.5 mM DTT in 25 mM Tris, pH 8.0. Lysates were separated from cell debris and remaining paraffin by centrifugation at 20.000 x *g* and manual removal of the top layer. This was followed by determination of the protein concentrations using BCA assay and subsequent sample preparation using an automated SDS-SP3 digestion and clean-up protocol[52,53]. Using a Bravo Automated Liquid Handling Platform (Agilent Technologies, Santa Clara, USA), the proteins were alkylated using iodoacetamide (IAA) for 30 minutes and blocked with an excess of 1,4-dithiothreitol (DTT). They were then bound to a 1:1 mixture of hydrophilic and hydrophobic magnetic beads at a high ACN (acetonitrile) concentration (>70%) and a beads to protein ratio of 10:1 by weight (10 μg beads:1 μg protein). After washing of the beads with 70% ethanol, the proteins were digested in solution in 50 mM HEPES (pH 8) with trypsin and LysC (enzyme-to-substrate ratio 1:50) overnight at 37 °C. The eluted peptides were then acidified using 100% formic acid (final concentration 1%) and desalted using AssayMAP tips on the Bravo robot. The final peptide concentration was determined by BCA assay.

Mass spectrometric data acquisition was performed on a Q Exactive HF-X instrument coupled to an easy nanoLC 1200 system (Thermo Scientific, Bremen, Germany). One microgram of peptides was injected per run and the separation was performed using an in-house packed reverse-phase column (20 cm, 1.9 μm beads, ReproSil Pur, Dr. Maisch GmbH) with a 110 min gradient from 3% to 60% (v/v) ACN, 0.1% (v/v) formic acid in water. The Q Exactive HF-X was operated in data-dependent mode with 60 K MS1 resolution=$3 \times 10^6$ ion count target and maximum injection time of 10 ms, followed by 20 MS2 scans with 45 K resolution, $1 \times 10^5$ ion count target, and maximum injection time of 86 ms.

Raw data were processed using the MaxQuant software Version 1.6.17.0 and the human reference proteome (UP000005640, downloaded 01/2019)[54]. For the database searches, Oxidation (M) and acetylation (N-term) were included as variable modifications; carbamidomethyl cysteine was included as a fixed modification. Peptides of a minimum length of seven amino acids were included in the search. The FDR was set to 0.01 for peptide and protein identifications. The Match-Between-Runs (MBR) feature was used for the analysis. We excluded proteins that were flagged by MaxQuant in the Reverse and Only identified by site column as well as proteins with less than three peptides. Contaminants of non-human proteins were identified manually and removed from the data set. After excluding samples with less than 1000 detected proteins, 51 specimens remained for further analysis. Further downstream analysis was performed using R Studio using base functions and the *lmfit* and *eBayes* function from the limma package to perform differential expression analysis for groupwise comparisons comparing the different tumor classes to normal sinonasal tissue[55]. Proteins with a $\log_2$ fold change >1.5 and an FDR < 0.05 were considered as differentially expressed. The list of significantly overexpressed genes in each tumor class compared to normal sinonasal tissue was subjected to overrepresentation analysis using the WebGestaltR package. All identified proteins that remained after filtering as described above were used as a reference gene list. Two gene sets of normal respiratory cell types were used to identify potential cells of origin for the respective tumor classes[22,23]. The WebGestaltR pipeline uses the Fisher's exact test to test for significance and the FDR method was used to adjust for multiple testing.

Functional analysis of proteomic data was performed using Cytoscape Version 3.9.1 and ClueGO Version 2.5.9. Differentially expressed proteins between the three proposed SNUC subtypes ACC, NEC-like IDH2 and NEC-like SMARCA4/ARID1A were used as input markers. Functional analysis of Reactome pathway terms was performed using a hypergeometric test, followed by Bonferroni step down correction. The results were visualized as functionally grouped networks using prefuse force directed layout.

T-SNE plots were generated as described above using a perplexity of 10 and 2000 iterations.

## DNA panel sequencing

The Ion AmpliSeq Library Kit 2.0 (Thermo Fisher Scientific) was used to perform library preparation of 10 ng of genomic DNA using the Ion AmpliSeq Cancer Hotspot Panel v2 (Thermo Fisher Scientific, catalog number 4475346). The final library was quantified with the Ion Library Quantitation Kit (Thermo Fisher Scientific). Samples were multiplexed and amplified on Ion Spheres Particles with Ion 540 Kit-Chef and were sequenced using Ion 540 Chip (Thermo Fisher Scientific) with an adapted standard protocol[56].

Selected samples were processed using the TruSight Oncology 500 Panel (Illumina, catalog number 20028214). 100 base pairs were sequenced in paired-end mode on an Illumina NextSeq 550 machine. The raw data was demultiplexed and analyzed using the TruSight Oncology 500 v2.2 Local App Docker. Briefly, demultiplexed reads were alignment to the GRCh37 (hg19) genome using the Burrows-

Wheeler Aligner, mapped reads were collapsed, re-aligned and stitched. Pisces software was used for somatic variant calling.

## DNA exome sequencing

DNA exome sequencing was performed at the CeGaT laboratory (Tübingen) using the Twist Human Core Exome Plus Kit (Twist Bioscience, catalog number 102027) on a NovaSeq 6000 sequencer (Illumina) to generate 2 × 100 bp reads. Sequence data were aligned to GRCh37 (hg19) genome using the Burrows-Wheeler-Aligner (Version 0.7.17)[57]. Somatic variants were called in comparison to matched normal tissue, requiring a coverage of at least 30 in both sequencings as well as an allele frequency of at least 0.05 in the tumor specimen. Tumor mutational burden was calculated using the parameters established for the Illumina TSO500 panel[58].

## Reporting summary

Further information on research design is available in the Nature Portfolio Reporting Summary linked to this article.

## Data availability

Raw DNA methylation data of all samples that have been collected for this study have been deposited in GEO under the accession GSE196228. Processed proteomics data are available at FigShare (https://doi.org/10.6084/m9.figshare.17144639). Due to privacy concerns, raw DNA sequencing and raw proteomics data cannot be made publicly available. Instead, this data has been deposited under controlled access in the European Genome Phenome Archive (EGA) under the accession numbers EGAS00001006712 (proteomics data; https://ega-archive.org/studies/EGAS00001006712) and EGAS00001006713 (DNA sequencing data; https://ega-archive.org/studies/EGAS00001006713). Requests for access should be addressed to the Data Access Committee of the Institute of Pathology LMU Munich (DAC@aimethylation.com). The time for response from the authors to applications will be within one month. All requests will be reviewed by the legal and data protection department of the LMU Munich. The following restrictions apply: (1) a data sharing agreement must be signed between the corresponding author and the data processor; (2) data will only be shared for scientific, non-commercial purposes; (3) the data processor must comply with the General Data Protection Regulation (GDPR) of the European Union, alternatively, they have to establish a data privacy policy that is adequate in the sense of the GDPR which will be assessed by the data protection department of the LMU Munich; (4) the data processor must delete all shared data after the investigation; (5) data must not be shared with any third party or individuals who are not authorized to access the data. For part of the study, publicly available data was retrieved from the TCGA database (https://www.cancer.gov/tcga). All other data is provided within the Supplementary Information and Supplementary Data files.

## Code availability

The code to reproduce the main analyses presented in this manuscript is available on FigShare (https://doi.org/10.6084/m9.figshare.17144639)[59].

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

## Acknowledgements

We gratefully acknowledge the excellent technical assistance of Peggy Wolkenstein, Ines Koch, Daniel Teichmann and Carola Geiler. We thank Thomas Cramer, Damian Rieke and Konrad Klinghammer for providing clinical information on single patients. Parts of Fig. 6 were created with BioRender.com. The results published here are in part based upon data generated by the TCGA Research Network: https://www.cancer.gov/tcga. This work was supported by the German Ministry of Education and Research (BMBF), as part of the National Research Node Mass spectrometry in Systems Medicine (MSCoresys), under grant agreement 031L0220B (to P.M.) and 031L0220A (to F.K.). Part of this study was further supported by the Deutsche Forschungsgemeinschaft (DFG – German Research Foundation) under grant agreement SFB 1449 Dynamic Hydrogels (projects C03, Z01; to P.M.). This study was in part further supported by the Berliner Krebsgesellschaft (JUFF201917; to P.J. and D.C.). Additional funding to D.C. was provided by the German Cancer Consortium (DKTK), partner site Berlin. P.J. was participant in the BIH-Charité Digital Clinician Scientist Program funded by the Charité – Universitätsmedizin Berlin, the Berlin Institute of Health (BIH) and the German Research Foundation (DFG) and is further supported by the Medical & Clinician Scientist Program (MCSP) of the LMU Munich. S.G. was awarded a medical doctoral research stipend for this project by the Berlin Institute of Health (BIH). K.R.M. gratefully acknowledges partial funding by the German Ministry for Education and Research under Grant 01IS14013A-E, Grant 01GQ1115, Grant 01GQ0850, as BIFOLD (ref. 01IS18025A and ref. 01IS18037A) and Patho234 (ref. 031L0207), the Institute of Information & Communications Technology Planning & Evaluation (IITP) grants funded by the Korea Government Grants 2017-0-00451 and 2019-0-00079.

## Author contributions

P.J., S.G., P.M., F.K. and D.C. designed the study. Samples were provided by C.D., A.J., S.S., S.S., H.B., P.H., B.E., S.F., J.H., W.P., M H., W.H., H.D., U.K., P.J., C.D., C.S., F.B., A.R., A.W., J.R.-I., S.P., C.I., L.C., R.D.M., A.M., U.S., J.L., V.J.L., M.F., M.L., S.L.-G., M.H., P.D.J., A.A., A.K., F.H., A.v.D., M.S., E.F., B.E.H., P.J. and D.C. DNA methylation analysis was performed

by L.H.M. and E.P.C. DNA sequencing was performed by I.H., C.V. and A.L. Proteomic analysis was done by R.R., C.F., R.F., M.H. and P.M. Computational analysis was performed by P.J., R.R., M.L., P.K., A.T., D.H., M.B., P.S. and K.R.M. P.J. and D.C. supervised the project. Resources were provided by M.H., S.P., D.TW.J., M.S., A.v.D., F.H. and D.C. Funding was acquired by P.J., K.R.M., P.M., F.K. and D.C. P.J., R.R., M.L., F.K. and D.C. wrote the original draft of the manuscript. All authors were involved in reviewing and editing of the final manuscript.

## Funding

## Competing interests

D.C. and A.v.D are listed as inventors on the patent application 'DNA-methylation based method for classifying tumor species' (PCT/EP2016/055337) filed by Deutsches Krebsforschungszentrum Stiftung des öffentlichen Rechts and Ruprecht-Karls-Universität Heidelberg. All other authors declare no conflicts of interest.

## Additional information

**Philipp Jurmeister**[1,2,3,4] ✉, **Stefanie Glöß**[5], **Renée Roller**[2,3,6], **Maximilian Leitheiser**[2], **Simone Schmid**[3,5], **Liliana H. Mochmann**[1], **Emma Payá Capilla**[1], **Rebecca Fritz**[2], **Carsten Dittmayer**[5], **Corinna Friedrich**[2,3,7,8], **Anne Thieme**[3,5], **Philipp Keyl**[2], **Armin Jarosch**[2], **Simon Schallenberg**[2], **Hendrik Bläker**[9], **Inga Hoffmann**[2], **Claudia Vollbrecht**[2,3], **Annika Lehmann**[2], **Michael Hummel**[2,3], **Daniel Heim**[2], **Mohamed Haji**[6], **Patrick Harter**[4,10,11], **Benjamin Englert** ®[12], **Stephan Frank** ®[13], **Jürgen Hench**[13], **Werner Paulus**[14], **Martin Hasselblatt**[14], **Wolfgang Hartmann** ®[15], **Hildegard Dohmen**[16], **Ursula Keber** ®[17], **Paul Jank** ®[18], **Carsten Denkert** ®[18], **Christine Stadelmann** ®[19], **Felix Bremmer**[20], **Annika Richter**[20], **Annika Wefers**[21,22,23], **Julika Ribbat-Idel**[24], **Sven Perner**[24,25,26], **Christian Idel**[27], **Lorenzo Chiariotti**[28,29], **Rosa Della Monica**[29], **Alfredo Marinelli**[30], **Ulrich Schüller** ®[23,31,32], **Michael Bockmayr** ®[2,31,32], **Jacklyn Liu**[33,34], **Valerie J. Lund**[33,34], **Martin Forster**[33,34], **Matt Lechner** ®[33,34], **Sara L. Lorenzo-Guerra**[35], **Mario Hermsen** ®[35], **Pascal D. Johann**[36], **Abbas Agaimy** ®[37], **Philipp Seegerer** ®[38], **Arend Koch**[5], **Frank Heppner** ®[3,5], **Stefan M. Pfister**[39,40,41], **David T. W. Jones**[39,42], **Martin Sill**[39], **Andreas von Deimling** ®[21,22], **Matija Snuderl**[43,44,45], **Klaus-Robert Müller**[38,46,47,48], **Erna Forgó**[49], **Brooke E. Howitt** ®[49], **Philipp Mertins** ®[6], **Frederick Klauschen**[1,4,48,50] & **David Capper** ®[3,5,50]

[1]Institute of Pathology, Ludwig Maximilians University Hospital Munich, Munich, Germany. [2]Institute of Pathology, Charité - Universitätsmedizin Berlin, corporate member of Freie Universität Berlin, Humboldt-Universität zu Berlin, Berlin, Germany. [3]German Cancer Consortium (DKTK), Partner Site Berlin, and German Cancer Research Center (DKFZ), Heidelberg, Germany. [4] German Cancer Consortium (DKTK), Partner Site Munich, and German Cancer Research Center (DKFZ), Heidelberg, Germany. [5]Charité – Universitätsmedizin Berlin, corporate member of Freie Universität Berlin and Humboldt-Universität zu Berlin, Department of Neuropathology, Charitéplatz 1, Berlin, Germany. [6]Proteomics Platform, Berlin Institute of Health (BIH) and Max Delbrück Center for Molecular Medicine in the Helmholtz Association (MDC), 13125 Berlin, Germany. [7]MDC Graduate School, Max Delbrück Center for Molecular Medicine in the Helmholtz Association, Berlin, Germany. [8]Humboldt Universität zu Berlin, Institute of Chemistry, Berlin, Germany. [9]Institute of Pathology, University Hospital Leipzig, Leipzig, Germany. [10]Institute of Neurology (Edinger Institute), Goethe-University Frankfurt am Main, Frankfurt am Main, Germany. [11]German Cancer Consortium (DKTK), Partner Site Frankfurt/Mainz, Frankfurt am Main, German Cancer Research Center (DKFZ), Heidelberg, Germany. [12]Institute of Neuropathology, Ludwig Maximilians University Hospital Munich, Munich, Germany. [13]Department of Neuropathology, Institute of Pathology, Basel University Hospital, Basel, Switzerland. [14]Institute of Neuropathology, University Hospital Münster, Münster, Germany. [15]Division of Translational Pathology, Gerhard-Domagk-Institute of Pathology, University Hospital Münster, Münster, Germany. [16]Institute of Neuropathology, University of Giessen, Giessen, Germany. [17]Institute of Neuropathology, Philipps-University, Marburg, Germany. [18]Institute of Pathology, Philipps-University Marburg and University Hospital Marburg, Marburg, Germany. [19]Institute for Neuropathology, University Medical Centre Göttingen, Göttingen, Germany. [20]Institute of Pathology, University Medical Center, Göttingen, Germany. [21]Department of Neuropathology, University Hospital Heidelberg, Heidelberg, Germany. [22]Clinical Cooperation Unit

Neuropathology, German Cancer Consortium (DKTK), German Cancer Research Center (DKFZ), Heidelberg, Germany. [23]Institute of Neuropathology, University Medical Center Hamburg-Eppendorf, Hamburg, Germany. [24]Institute of Pathology, University of Luebeck and University Hospital Schleswig-Holstein, Campus Luebeck, Luebeck, Germany. [25]Pathology, Research Center Borstel, Leibniz Lung Center, Borstel, Germany. [26]German Center for Lung Research (DZL), Partner Site Luebeck, Luebeck, Germany. [27]Department of Otorhinolaryngology, University Hospital Schleswig-Holstein, Campus Lübeck, Lübeck, Germany. [28]Dipartimento di Medicina Molecolare e Biotecnologie Mediche, Università degli Studi di Napoli Federico II, Via S. Pansini 5, 80131 Naples, Italy. [29]CEINGE Biotecnologie Avanzate, 80145 Naples, Italy. [30]Department of Medicina Clinica e Chirurgia, University Federico II, Naples, Italy. [31]Department of Pediatric Hematology and Oncology, University Medical Center Hamburg-Eppendorf, Hamburg, Germany. [32]Research Institute Children's Cancer Center Hamburg, Hamburg, Germany. [33]UCL Cancer Institute, University College London, 72 Huntley Street, London WC1E 6BT, UK. [34]UCL Academic Head and Neck Centre, Division of Surgery and Interventional Science, University College London, London, UK. [35]Department of Head and Neck Oncology, Instituto de Investigación Sanitaria del Principado de Asturias (ISPA), Oviedo, Spain. [36]Swabian Childrens' Cancer Center, University Childrens' Hospital Augsburg and EU-RHAB Registry, Augsburg, Germany. [37]Institute of Pathology, Friedrich-Alexander-University Erlangen-Nürnberg, University Hospital, Erlangen, Germany. [38]Machine-Learning Group, Department of Software Engineering and Theoretical Computer Science, Technical University of Berlin, Berlin, Germany. [39]Hopp Children's Cancer Center Heidelberg (KiTZ), Heidelberg, Germany. [40]Division of Pediatric Neurooncology, German Cancer Research Center (DKFZ) and German Cancer Consortium (DKTK), Heidelberg, Germany. [41]Department of Hematology and Oncology, Heidelberg University Hospital, Heidelberg, Germany. [42]Division of Pediatric Glioma Research, German Cancer Research Center (DKFZ), Heidelberg, Germany. [43]Division of Neuropathology, NYU Langone Health, New York, USA. [44]Laura and Isaac Perlmutter Cancer Center, NYU Langone Health, New York, USA. [45]Division of Molecular Pathology and Diagnostics, NYU Langone Health, New York, USA. [46]Department of Artificial Intelligence, Korea University, Seoul, South Korea. [47]Max-Planck-Institute for Informatics, Saarbrücken, Germany. [48]BIFOLD – Berlin Institute for the Foundations of Learning and Data, Berlin, Germany. [49]Stanford University School of Medicine, Stanford, CA, USA. [50]These authors contributed equally: Frederick Klauschen, David Capper. ✉e-mail: philipp.jurmeister@med.uni-muenchen.de

