## [Peer Review File · Nature Communications]

REVIEWER COMMENTS

Reviewer #1 (Remarks to the Author): expertise in WES and proteomics analysis and DNA methylation-based machine learning classification models

In this study, the authors used DNA methylation data to comprehensively investigate the molecular patterns of sinonasal undifferentiated carcinomas (SNUCs), which were poorly defined and disputed in previous studies. According to DNA methylation, SNUCs were assigned to four distinct molecular classes harboring other different molecular features, like mutations, copy number variations and protein expressions. Overall, this work is interesting and the ms is well written. But there are some concerns that should be addressed.

- (1) TCGA mostly used 450K or 27K, while the authors' dataset mostly used EPIC array. Are there any batch effects in the classification?
- (2) The analyses of copy number variations in different subtypes showed very distinct patterns (Supplementary Figure S3). Will copy number variations be superior to DNA methylation in terms of classification of subtypes and non-sinonasal tumors?
- (3) There have been several similar studies on the subtyping of SNUCs based on DNA methylation (References 8-11), it is necessary to include further discussions about what new knowledge readers could obtain from this study compared with Ref. 8-11.
- (4) There are no legends for supplementary figures.
- (5) For figure 5C, the accuracy is ambiguous. Classification accuracy is a metric that summarizes the performance of a classification model as the number of correct predictions divided by the total number of predictions. How could accuracy achieve 1 with a sensitivity of 0.904.
- (6) a typo in line 270 "and".

Reviewer #2 (Remarks to the Author): expertise in multi-omics, metabolomics and proteomics

Jurmeister et al. report the result of DNA methylation analysis and classification of tumor specimen into types of sinonasal tumors.

Overall, the authors provide detailed information and molecular characterization of sinonasal tumors. The technical approach seems to be state-of-the-art, and relevant high-quality metadata and clinical classifications are integrated in the discussion. The manuscript is well-written and in principle easy to follow.

Although a critical evaluation of the author's findings is clearly needed before they can be translated into clinical decision, as is noted by the authors, their findings could serve as a basis for implementing more informative diagnostic markers, predicting known treatment strategies that could be repurposed for specific subtypes, or for developing new treatments. The authors' discussion and promotion of outlier detection and other practical considerations is appreciated. As such, the author's finding will likely be of high interest and even relatively immediate value for experts in the field and/or clinicians.

The authors generate a lot of data that also seems to be informative with regards to their question. However, to allow a broader readership to fully appreciate the extent of their advance, some clarifications and a better contextualization with respect to existing approaches are needed.

General comments (in more detail below):

- (a) Please better explain the rationale for seeking to stratifying tumors by methylation signature. Is it merely for practical reasons (availability of technology, cost, ...) or does it reveal biological information that is not captured by alternative classification approaches (e.g. the mentioned WHO classification)?
- (b) While the goal of this study was clearly to stratify tumor subtypes, in general it is not very clear what – aside from said differentiation – the authors have learnt specifically from the DNA methylation profiles. E.g., whether the authors have gained any mechanistic evidence for the differences in clinical course between the different subtypes, or think it would be possible to derive such information from their data.
- (c) It is left somewhat unclear why two different classification approaches were used (first t-SNE and later machine learning), and whether they were connected or present independent solutions. It appears that the ML-based approach is "re-inventing" the earlier t-SNE based solution for

classification. See also more detailed comments below.

Specific comments and suggestions:

- (1) Line 173: How is a "stable" epigenetic class defined and how is it measured here?
- (2) In the presentation of the initial classification results using t-SNE (line 218 onwards) it is somewhat unclear which characteristics that the authors describe originate from the DNA methylation data. The way it is written now, it appears like complementary measurements like histological features or mutational profiling were actually more informative than the methylation patterns. In particular in the description of the ACC class (line 238f) it seems that no results from methylation analysis are even mentioned. Since the classification algorithm did separate the clusters by the DNA methylation data, and this is the major claim of the paper, it seems that the underlying characteristic methylation signatures should be the center piece of this section. If not, please justify and/or also describe additional measurements performed at the beginning of the section.
- (3) Perhaps to help in the above point, I would strongly recommend including a brief introduction to the key features that DNA methylation profiling measured and what conclusions can be drawn from them to appeal to a broader audience. In particular, were copy number profiles derived from DNA methylation data, or were additional measurements done?
- (4) How does the molecular information gained by analyzing DNA methylation differ from the information used so far for classifying sinonasal tumors, e.g. what is the WHO classification of Head and Neck tumors that the authors mention based on? Was it obvious that most classes would be consistent between the standard classification and one based on methylation patterns? Why does the standard classification not, or why does epigenetics better, define the SNUC subtypes?
- (5) The authors then include MS-based proteomics measurements. A brief note as to the rationale and expectation from proteomics measurements at the beginning of the section would be appreciated.
- (6) The figure captions are somewhat sparse, in particular but not only in Figure 3. Description of panel A should include details on the clustering performed on the heatmap data. In panel B statistical tests used in the differential analysis must be mentioned (the y-axis label is also missing in the figure itself). There are duplicate descriptions of panel C with different contents, please resolve and describe the statistical comparisons. Panel D please describe which tissues are shown etc. Panel E is missing a description of several items like the enrichment analysis, statistics and FDR, and an explanation how similarity is measured.
- (7) In line 288f the authors seem to attempt to cluster samples based on proteomics measurements, similar to what was done on the methylation data. Only here, the authors do it via hierarchical clustering and visual inspection in the heatmap. If the authors wish to compare the clustering, the same method should be used (i.e. t-SNE).
- (8) Line 311f please briefly describe the premise/idea behind the overrepresentation analysis (overrepresentation of what against which reference? how was it tested statistically, hypergeometric test?), since it is also not explained in the methods section.
- (9) Next, the authors include additional mutational profiling data. In general, it is unclear why the authors sought to generate even more data, what information was missing? Is there a reason why this analysis was only done for one of the new classes? At the end of the paragraph it is also unclear what the authors conclude with regards to the overall question in the paper.
- (10) Lines 329-330, please avoid drawing conclusions from single observations (n=1).
- (11) Finally, the authors implement the methylation-based classification in a machine learning framework. While the authors point out practical considerations for diagnostic use, it seems that the ML algorithm makes the earlier t-SNE analysis obsolete and is "re-inventing" the classification. This seems confusing given that most of the new findings of the paper were generated with the t-SNE based classification, and it is not clear whether or how that knowledge was used in the ML framework. E.g. was the t-SNE based classification as a training input in developing the ML classifier?
- (12) The performance evaluation is somewhat hard to follow, again because it is not clear whether this implemented the earlier t-SNE classification, or what the overlap of the two classification approaches is here. E.g. were the reclassifications pointed out in lines 392f and 397f consistent with the t-SNE analysis? If not, was it expected and what are the general differences in

performance one should expect with the two approaches?

Reviewer #3 (Remarks to the Author): expertise in sinonasal tumours and subtypes

This is a molecular analysis of a retrospective, multicenter, large sample size cohort of sinonasal tumor biospecimens utilizing DNA methylation and differences in expression as a means to improve histopathologic diagnosis. A secondary aim is to provide additional clarity into sinonasal undifferentiated carcinoma (SNUC) as a diagnostic entity, utilizing cluster analysis (DNA methylation and mass spectroscopy proteomics) to elucidate specific molecular differences within this group which often challenges pathologists. Finally, a machine learning algorithm based on DNA methylation was incorporated to further refine diagnostic accuracy. The authors in general were able to demonstrate clear differences in methylation patterns which correlated with distinct diagnoses, as well as to create 4 new clusters within the SNUC cohort based on these differences.

As sinonasal tumors are extremely rare and very challenging to study, with frequent misdiagnosis, the authors are to be applauded for this multi-institutional effort which provides some new insights into SNUC as an entity and sheds light into the utility of DNA methylation as a tool in this endeavor. The Figures, Tables, and supplemental data are illustrative. There is no concern regarding word count and references.

However, because the study revolves around previously made diagnoses from various centers and uses new molecular data for correlation, the fundamental question is whether or not the reclassified tumors were actually misdiagnosed to begin with. Also, as the WHO classification for head and neck tumors was updated in 2017, how many of these samples were diagnosed before that time when modern, more updated nomenclature was not in use?

Major Comments:

- What is "normal sinonasal tissue" - where was it collected from, and was each specimen confirmed to be free of tumor? In how many cases was this tissue adjacent to tumor?
- Why were so many neuroendocrine carcinomas (11/24) and SNUCs (15/84) excluded due to noise? How does this impact the results?
- Were histopathologic diagnoses provided locally at each contributing center or were they re-reviewed centrally for this study?
- The SMARCB1 altered cluster is well established as a subtype of SNUC, especially increasingly recognized in the last few years. How many tumors in this cohort were classified broadly as SNUC without consideration of this specific subtype, or before recognition of this entity?
- Regarding the SNUC misclassified ACC cluster - were the 25 cases from multiple centers or there was a predominance of origin from one center? To state that "tumors that represent previously misclassified adenoid cystic carcinomas are highly aggressive" may be an overstatement, as they should technically follow the natural biological behavior of ACC as opposed to SNUC.
- The results from comparing survival based on class itself are not generalizable without more detailed patient data (e.g., tumor extent, metastases, treatments rendered), and thus should be interpreted with caution.
- Did the authors compare survival between SNUC ACC and the diagnosed ACC cohorts?
- What is the source of the 8,065 tumor and normal samples?
- Regarding the 34 cases excluded in the cluster analysis, why do the authors think they were incompatible? The comment regarding these being rare diagnoses does not apply and is an inadequate explanation, as only a minority of the noise points are singular diagnoses.

Minor Comments:

- WHO classification requires citation.
- H-score description requires a citation.
- Please relabel Conclusions as Discussion section.

Formatting Comments:

- Some of the interpretation of results which should be in the Discussion section are directly written in the Results section; this should be separated accordingly.
- Similarly, some information in the Results section should belong in the Methods section.
- Many statements are made without reference support or based on anecdotal evidence ("Early clinical data suggests that these patients might benefit from treatment with AURKA, EZH2 or PD1 inhibitors."). These statements should be avoided overall given lack of scientific basis.
- Please proofread the manuscript and figures to address all typos.
- Parts of the manuscript are written in informal language and should more reflect formal scientific writing.

Reviewer #4 (Remarks to the Author): expert in proteomics

This is an impressive study looking at methylation patterns of a diverse population of sinonasal cancers. While many existing subtypes are recapitulated from methylation subgrouping, the sinonasal undifferentiated carcinomas (SNUCs) were grouped into more identifiable subgroups. Subsequent molecular analysis identified gene mutation or loss as key diagnostic indicators of subgroup, thereby identifying molecular heterogeneity within the SNUC histology. Further, the authors developed a machine learning signature to predict their molecular subtyping from methylation data thereby offering a methylation-based molecular pathology approach to identifying SNUC subtypes. This work may drive the reclassification of a relatively uncertain category of sinonasal cancers, leading to improvements in diagnosis of the disease as well as potential new therapeutic options which can be derived from these additional insights.

Overall this study is well designed and executed to understand the methylation-driven subtyping of sinonasal cancers.

Several general comments:

1. There are several sections of the methodology which are not completely clear and could be improved. These are noted in the specific comments below. The methodology is well designed and applied – the comment is more in the description of the techniques.
2. This paper is focused on diagnostic accuracy and thus in both the methylation analysis and proteomics analysis, molecular comparisons (i.e. differential analysis followed by enrichment) were not performed across putative subtypes. Given the discovery of methylation subtypes (driven by differences in methylation sites in specific genes), understanding the methylation-specific differences between subtypes could better elucidate the biological differences between tumor subclasses. In the case of the proteomics experiment, comparisons were made of the tumor subclass vs. normal tissue. While identifying markers differing from normal tissue is important for pathology diagnosis, there is significant value in identifying the broader molecular characteristics that differ between types. This is particularly true in the subgroups (ACC and NEC-like SMARCA4/ARID1A) where a clear driver is not readily apparent.

Specific comments are listed below.

- The first paragraph of the results (Identification of DNA methylation ...) is not completely clear. The first tsne/DBSCAN combination determined outlier samples. Once outlier samples were removed from the cohort, was the tsne re-applied and DBSCAN used to determine the number of clusters/groups? It is not clear if the tsne/DBSCAN clustering was used from the filtering step or was rebuilt once outliers were removed. It seems that rebuilding the model excluding outliers would be more robust, so perhaps this was done.
- Second para of results: "Iterative random down-sampling and correlation analysis" seems like it would be more understandable as "Iterative random down-sampling with correlation analysis". That is, the two methods are tied together to assess stability of the tsne analysis (not necessarily the classes/clusters).
- Second para: It seems that the sensitivity/specificity of methylation profiles for the non-SNUC related groups is 1/1. Is this correct? It may be worth mentioning this point explicitly. The clustering uniquely separated these other tumor types perfectly (if that is the case).
- The authors should be commended for examining potential batch effects. One further variable to consider could be the study (from Supplemental Table 3). It does not appear to be a confounder,

but perhaps the authors have tested this variable.

- Figure 3A: The legend shows that the data is centered log₂ intensity. It is not clear if this means that the data is centered by protein. The heatmap does not appear to have centered rows as many rows are primarily one color.

- Figure 3B: It would be important to label these axes. Is the x axis log₂ or log₁₀ fold changes?

- A number of sections (such as the proteomics section) do not contain many quantitative results. For instance, some of the enrichment findings are described but they do not have the corresponding p values in the text. The figures provide a summary of the statistical results, but specific findings could be more precisely reported if the p values (or FDR) are included.

- Figure 5D: Including the AUC can provide a numerical summary of the curve and further emphasize the high sensitivity/specificity of the classifier. Also, it is not clear what data the ROC is derived from or what thresholds are varied to generate the ROC points.

Point-by-point response

Reviewer comments are shown in black.

Author responses are shown in blue.

Changes in the revised manuscript are highlighted in yellow.

Reviewer 1

Comment 1: “TCGA mostly used 450K or 27K, while the authors’ dataset mostly used EPIC array. Are there any batch effects in the classification?”

Response: We thank the reviewer for bringing up this important technical aspect. In this study, we only used methylation data that was generated using the 450k or EPIC BeadArrays generation. The inclusion of 27k arrays would greatly reduce the number of CpG sites available for classification while only offering little benefit since the number of publicly available 27k datasets is comparably low and the generation is no longer being produced or used. All included samples from the TCGA are part of the non-sinonasal cohort and thus only used for the binary outlier detection. However, we would like to extend the reviewer’s comment slightly and elaborate on the potential batch effects with regards to the differences in array type (450k v. EPIC) in general, including the sinonasal classification.

The array type distribution across the different cohorts is shown in Response Letter Table 1. A majority of cases in the sinonasal reference set were profiled using the EPIC array (283/395; 72%). These cases have been collected over the course of several years and the data from some entities were derived from previous studies that specifically focused on certain diagnoses. The sinonasal test set solely consists of EPIC-type samples, as they all have been measured recently and the EPIC array design is the current commercially available generation. The non-sinonasal cohort, including the TCGA samples mentioned in the reviewer’s comment, mainly consists of 450k-type samples (6,541/8,063; 81%).

Cohort	Total	450k	EPIC
Sinonasal reference cohort	395	112 (28%)	283 (72%)
Sinonasal test cohort	53	0 (0%)	53 (100%)
Non-sinonasal cohort*	8,063	6,541 (81%)	1,522 (19%)

Response Letter Table 1: Chip design distribution across cohorts. *includes TCGA samples

We would expect strong batch effects to be apparent in unsupervised visualization methods such as a t-SNE, as affected cases usually aggregate in a separate group (if the batch effect exceeds the biological differences) or in a shared area of a group defined by biological differences. As shown in Supplementary Figure S2D, there is no apparent array type-related batch effect within the sinonasal reference cohort.

The sinonasal classification also shows no evidence for such batch effect, as all considered samples are classified correctly. This is remarkable, since there are sinonasal entities that are predominantly or even exclusively present as 450k samples in the training set (10/10 pituitary adenoma, 19/19 craniopharyngioma, 10/12 Ewing’s sarcoma, 13/56 olfactory neuroblastoma) but the considered EPIC samples of these classes in the test set (8 pituitary adenoma, 1 craniopharyngioma, 2 Ewing’s sarcoma, 8 olfactory neuroblastoma) are all classified correctly. Thus, the classifier seems to be able to differentiate the sinonasal classes successfully, regardless of the chip designs that were presented during its training.

As there are no 450k samples in the sinonasal test set, the sensitivity of the outlier detection task cannot be compared across chip types. Due to the over-representation of 450k samples in the non-sinonasal cohort, the sinonasal entities with mostly 450k samples in the training set mentioned above could be expected to be more likely (incorrectly) classified as ‘unknown’. However, this is not the case: Of these samples, all but one Ewing’s sarcoma sample are recognized as a sinonasal class, resulting in a sensitivity of 95% (19/20) on this set, which is higher than the overall sensitivity of 90.4% (47/52) on the test set, contradicting the potential batch effect described here.

Comment 2: “The analyses of copy number variations in different subtypes showed very distinct patterns (Supplementary Figure S3). Will copy number variations are superior to DNA methylation in terms of classification of subtypes and non-sinonasal tumors?”

Response: We agree with the reviewer that the frequency of different copy number alterations differs between most tumor classes. However, it is also apparent from the Supplementary Figure S3 that there are many classes without highly recurrent copy number changes. Members of our group actually investigated the specificity of different omics data for tumor classification based on the TCGA cohort in a previous study (Heim et al. *Genome Medicine* 2018). They reported that somatic mutations as well as copy number profiles showed a considerable overlap between different tumor entities and that mRNA expression, proteomics and particularly DNA methylation data is much more suitable for tumor classification.

Following the reviewer’s point, we have extracted the raw copy number data for each probe. Unsupervised t-SNE analysis (Response Letter Figure 1) showed a very poor separation of the different histological classes.

Response Letter Figure 1: t-SNE analysis based on copy number data shows poor separation between the different tumor classes.

Thus, we conclude that copy number data can provide additional confidence for the DNA methylation-based classification if the profiles are in line with the most frequently observed alterations in the respective class. However, it is clearly inferior to DNA methylation. We have included a short discussion on this aspect in the revised Discussion section of our manuscript (p. 17): “Copy number profiles derived from DNA methylation data revealed different recurrent copy number alterations between the tumor classes. This information may provide additional confidence to the DNA methylation-based classification, although the sensitivity and specificity of these alterations for classificatory purposes seem limited.”

Comment 3: “There have been several similar studies on the subtyping of SNUCs based on DNA methylation (References 8-11), it is necessary to include further discussions about what new knowledge readers could obtain from this study compared with Ref. 8-11.”

Response: DNA methylation analysis of sinonasal tumors has indeed been attempted by two previous studies, namely Dogan et al. *Modern Pathology* 2019 and Capper et al. *Acta Neuropathologica* 2018. However, both studies only covered a narrow spectrum of sinonasal cancers with rather limited total case numbers. The current study has included a variety of additional entities, expanding the cohort to the whole spectrum of diagnostically relevant tumor classes and has clearly increased the total numbers of samples. This allowed for the first time to identify new DNA methylation based tumor classes among sinonasal tumors. Our current study further performed additional molecular analyses (NGS, proteomics) of these new classes to define typical molecular alterations for these classes. Our data now is dense enough to develop DNA methylation-based classification tools for sinonasal tumors.

The main difference between these two studies and our current work is the actual diagnostic relevance. Our study includes almost the whole spectrum of tumors that can occur in the sinonasal region while the two previous publications focused on SNUCs and olfactory neuroblastomas, respectively. Furthermore, neither of these studies provided a ready-to-use machine learning algorithm to actually assist histopathological diagnosis. In addition, our study is the first to report proteomic profiling and exome sequencing on these tumor types, providing additional insights into the biology of these tumors. To address the reviewer’s point, we have highlighted the main advances of our study in comparison to previous work in the Discussion section of the updated manuscript (pp. 16 & 17).

Comment 4: “There are no legends for supplementary figures.”

Response: We apologize that the legends for the Supplementary Figures were not provided correctly. For the revision, we made sure that this data is included and properly uploaded to the editorial system.

Comment 5: “For figure 5C, the accuracy is ambiguous. Classification accuracy is a metric that summarizes the performance of a classification model as the number of correct predictions divided by the total number of predictions. How could accuracy achieve 1 with a sensitivity of 0.904 “

Response: The specificity and sensitivity reported in our manuscript refers to the binary classification of the outlier detection, i.e. ‘sinonasal’ versus ‘Unknown’. The accuracy on the other hand refers to the assignment of sinonasal tumors that have not been excluded by the outlier detection as ‘Unknown’. We already described this approach in the Material and Methods section but agree with the reviewer that the current terminology might be ambiguous for the reader. To emphasize that these terms refer to the outlier detection and sinonasal tumor classification, respectively, we have chosen to use the terms ‘outlier detection specificity/sensitivity’ and ‘sinonasal accuracy’ in Figure 5C and added a more detailed description in the Materials and Methods section (p. 25). The labels in Figure 5 were adjusted accordingly.

Comment 6: “a typo in line 270 “and”.”

Response: We thank the reviewer for making us aware of this error which has been corrected in the updated version of the manuscript (p. 10).

Reviewer 2

Comment 1: “Please better explain the rationale for seeking to stratifying tumors by methylation signature. Is it merely for practical reasons (availability of technology, cost, ...) or does it reveal biological information that is not captured by alternative classification approaches (e.g. the mentioned WHO classification)?”

Response: We apologize for not making this aspect clear enough. DNA methylation plays a significant role in the differentiation of cell types and numerous studies have shown that global DNA methylation signatures are highly tissue-specific (e.g., Brena et al. *Nature Genetics* 2006). Although epigenetic alterations are one of the hallmarks of cancer development, valuable information about the cell of origin is usually retained in tumor cells. This makes DNA methylation an optimal tool for tumor classification. Our group and others have conducted various studies showing the feasibility of this approach (e.g., Capper et al. *Nature* 2018, Jurmeister et al. *Science Translational Medicine* 2019). In fact, DNA

methylation profiling has recently been introduced into the latest revision of the WHO Classification of Central Nervous System Tumors, which recognized that some novel entities (e.g., high-grade astrocytoma with piloid features) can currently only be identified by their specific DNA methylation profile. However, this is a novelty and still the exception, as the definition of different tumor entities in the WHO classification is usually mainly based on conventional histomorphology and immunohistochemical features. However, there can be a substantial overlap of histopathological patterns between different tumor entities. Furthermore, common defining features can get lost during dedifferentiation. In recent years, the WHO classification started to include mutations and fusion events for the definition of certain tumor entities. However, these alterations are usually not specific for a single entity and in most cases, they do not occur in all tumors of the same class. Furthermore, different tumor entities are not necessarily characterized by specific alterations. Therefore, the value of these methods for diagnostic purposes is somewhat limited.

As mentioned by the reviewer, there are also technical aspects that make DNA methylation an attractive tool for tumor classification: methylated DNA is highly robust (in contrast to e.g., RNA), enabling the retrospective analysis of FFPE samples, almost irrespective of sample age. Using this approach, significant cohorts of even extraordinarily rare cancers can be assembled, as shown in this study.

In line with the reviewer's comment, we have revised the Introduction section of our updated manuscript file to better explain the rationale for choosing DNA methylation profiling for tumor classification (p. 4).

Comment 2: "While the goal of this study was clearly to stratify tumor subtypes, in general it is not very clear what – aside from said differentiation – the authors have learnt specifically from the DNA methylation profiles. E.g., whether the authors have gained any mechanistic evidence for the differences in clinical course between the different subtypes, or think it would be possible to derive such information from their data."

Response: The main goal of this research project was to generate an objective classification tool for tumors that are extremely hard to classify for pathologists. Additional focus was set on identification of subclasses within sinonasal undifferentiated carcinomas and molecular and clinical characterization of these subclasses.

The mechanistic interpretation of differences between the DNA methylation signatures of different classes is not straightforward and we further discuss this aspect in our response to Comment 5. In contrast to DNA methylation, proteomic data is much more useful to provide functional insights. Although the primary focus of our study was the application of an innovative classification technique, we have now extended on the functional analysis of the proteomics data. This is further discussed in our response to Comment 1 by Reviewer 4.

Comment 3: "It is left somewhat unclear why two different classification approaches were used (first t-SNE and later machine learning), and whether they were connected or present independent solutions. It appears that the ML-based approach is "re-inventing" the earlier t-SNE based solution for classification. See also more detailed comments below."

Response: We thank the reviewer for the constructive feedback and apologize for not making this point clear enough. It is important to distinguish t-SNE as a dimensionality reduction tool from true machine learning classifiers. We used t-SNE and the DBSCAN algorithm to define epigenetic classes in an unsupervised fashion. These classes were then molecularly and clinically characterized. Finally these new classes were used as reference data for the training of an ML-based classifier. T-SNE is not suitable for the classification of new samples for various reasons. First, the location of individual points within the t-SNE plot can differ over different iterations and is depending on the selected parameters and the composition of the reference cohort. Furthermore, t-SNE does not allow the implementation of confidence metrics or outlier detection methods. All this can be achieved using a dedicated machine learning classifier. We have revised our manuscript (p. 13) as well as Supplementary Figure S1 to make this aspect clearer.

Comment 4: "Line 173: How is a "stable" epigenetic class defined and how is it measured here?"

Response: As described in the Methods section, the DBSCAN method was used to define epigenetic classes and to assign cases to their individual classes. This technique is able to identify 'noise' points which do not stably correspond to one of the defined classes. The stability of the classes was determined by iterative random down-sampling to 80% of the total cohort, as previously described (Capper et al. *Nature* 2018). We calculated the Pearson's correlation coefficient of the x and y coordinates for all samples after 300 iterations. As described in the Results section, the results showed a high stability for the previously defined classes with a median class correlation coefficient of 0.992 (range: 0.945-0.999).

Comment 5: “In the presentation of the initial classification results using t-SNE (line 218 onwards) it is somewhat unclear which characteristics that the authors describe originate from the DNA methylation data. The way it is written now, it appears like complementary measurements like histological features or mutational profiling were actually more informative than the methylation patterns. In particular in the description of the ACC class (line 238f) it seems that no results from methylation analysis are even mentioned. Since the classification algorithm did separate the clusters by the DNA methylation data, and this is the major claim of the paper, it seems that the underlying characteristic methylation signatures should be the centerpiece of this section. If not, please justify and/or also describe additional measurements performed at the beginning of the section. Perhaps to help in the above point, I would strongly recommend including a brief introduction to the key features that DNA methylation profiling measured and what conclusions can be drawn from them to appeal to a broader audience.”

Response: As mentioned by the reviewer, the DNA methylation data was used for the underlying definition of these classes. We have revised the introduction of the “Reassessment of SNUC classes”, (p. 7) which now read “[...] the four SNUC classes defined by distinct DNA methylation profiles [...]” to make this aspect clearer.

As described in the beginning of the paragraph the reviewer refers to, we carefully examined the additionally available histopathological and molecular data of these cases. As described in their respective paragraphs, we observed enrichment of specific features in the four undifferentiated sinonasal tumor classes: *IDH2* hotspot mutations in the NEC-like *IDH2* class, *MYB* fusions and subtle histological adenoid cystic features in ACC class tumors, loss of *SMARCB1* with subsequent loss of *INI1* expression in the *SMARCB1* class as well as neuroendocrine features in the NEC-like *SMARCA4/ARID1A* class. While DNA methylation is a robust classification tool, the underlying signatures are typically rather uninformative, scattered across the whole genome and mainly located in the gene body region with uncertain association with gene expression. This can also be seen in the distribution of the 20,000 CpG sites that were selected for the machine learning classifier (see Response Letter Figure 2). The distribution is comparable to the overall array design, there is no enrichment of functionally relevant promoter regions. This can also be seen in other DNA methylation-based classifiers. We have added this observation to the Results (p. 5) and Discussion section (p.) and have included the figure shown below as a new Supplementary Figure S1.

Response Letter Figure 2 (new Supplementary Figure S1):

A Distribution of the 20,000 CpG site used by the classifier across different gene region categories compared to the 450k array design. The overall distribution is comparable and there is no enrichment in functionally relevant promoter regions.

B Karyoplot of chromosome 1 to 22 showing the density distribution per megabase of all CpGs from the 450k array design and the 20,000 most variant CpGs that were able to separate the sinonasal tumor classes. There is an overall very similar distribution without clear enrichment in specific chromosomal regions.

Comment 6: “In particular, were copy number profiles derived from DNA methylation data, or were additional measurements done?”

Response: As described in the Method section (p. 25), all copy number profiles were derived from DNA methylation data. We now also mention this aspect in the Results section of the revised manuscript file, to make this clearer (p. 8). The respective paragraph now reads: “Copy number profiles derived from DNA methylation data showed highly recurrent chromosomal aberrations [...]”.

Comment 7: “The authors then include MS-based proteomics measurements. A brief note as to the rationale and expectation from proteomics measurements at the beginning of the section would be appreciated.”

Response: We decided to perform proteomic profiling to evaluate if cases from the SNUC classes were characterized by specific protein expression profiles. We intended to use this information to identify markers that could be used for histopathological diagnosis if DNA methylation profiling is not feasible or available, as well as to obtain insight on potential cells of origin of the new tumor subclasses. We were able to show that a combination of KRT18 and UCHL1 is helpful to differentiate the different classes from each other and important histological mimics. Furthermore, we were able to propose potential cells of origin using previously defined gene sets. The following text has been added to the beginning of the respective paragraph (p. 9): “To identify characteristic protein expression profiles and potential cells of

origins for specimens from the four SNUC classes, we performed mass spectrometry-based proteomics.”

Comment 8: “The figure captions are somewhat sparse, in particular but not only in Figure 3. Description of panel A should include details on the clustering performed on the heatmap data. In panel B statistical tests used in the differential analysis must be mentioned (the y-axis label is also missing in the figure itself). There are duplicate descriptions of panel C with different contents, please resolve and describe the statistical comparisons. Panel D please describe which tissues are shown etc. Panel E is missing a description of several items like the enrichment analysis, statistics and FDR, and an explanation how similarity is measured.”

Response: We thank the reviewer for the constructive feedback. We have reviewed all figure legends and extended them appropriately. With regards to Figure 3, we have replaced the initial heatmap with a t-SNE plot as suggested in the next comment. We have added the missing axis labels and removed the duplicate description of panel C.

Comment 9: “In line 288f the authors seem to attempt to cluster samples based on proteomics measurements, similar to what was done on the methylation data. Only here, the authors do it via hierarchical clustering and visual inspection in the heatmap. If the authors wish to compare the clustering, the same method should be used (i.e. t-SNE).”

Response: We agree with the reviewer that it is more consistent to also use t-SNE to estimate the similarity between the proteomic specimens. Figure 3 has been adjusted accordingly.

Comment 10: “Line 311f please briefly describe the premise/idea behind the overrepresentation analysis (overrepresentation of what against which reference? how was it tested statistically, hypergeometric test?), since it is also not explained in the methods section.”

Response: We apologize for not making this point clear enough. Overrepresentation analysis was performed to compare the general differentiation of the molecular tumor classes and to test for similarities with potential cells of origin. We have revised this paragraph accordingly (p. 11), which now reads: “To identify potential cells of origins, differentially expressed proteins from all tumor classes in comparison to normal sinonasal tissue were subjected to overrepresentation analysis [...]”. Differential expression analysis was performed in comparison to normal sinonasal tissue as a reference. Significantly expressed proteins were subjected to overrepresentation analysis using the WebGestalt pipeline which uses a Fisher’s exact test to test for significantly enriched genes in given gene sets. We now provide a more detailed description of this analysis in the Materials and Methods section of the updated manuscript file (pp. 26-27).

Comment 11: “Next, the authors include additional mutational profiling data. In general, it is unclear why the authors sought to generate even more data, what information was missing? Is there a reason why this analysis was only done for one of the new classes? At the end of the paragraph it is also unclear what the authors conclude with regards to the overall question in the paper.”

Response: The rationale to perform additional mutational analysis was to identify potential driver mutations for tumors from the newly identified subtype of sinonasal undifferentiated carcinoma that has not been identified as a distinct tumor class before (called NEC-like SMARCA4/ARID1A class because of the molecular results) In contrast to the other three classes that all harbour recurrent molecular driver mutations known from the literature (NEC-like IDH2 (IDH2 R172 mutations), SMARCB1 (deletion of the *SMARCB1* gene locus) and ACC (MYB/MYL fusions)), a recurrent driver for the NEC-like SMARCA4/ARID1A class was unknown. We have rewritten the introduction of this paragraph to address the reviewer’s point (p. 12), the sentence now reads: “As a clear driver for the new NEC-like SMARCA4/ARID1A molecular tumor class was not apparent in the available retrospective data [...]”.

Comment 12: “Lines 329-330, please avoid drawing conclusions from single observations (n=1).”

Response: We agree with the reviewer that this statement has been overambitious and we have rephrased the sentence accordingly (p. 18): “This also included one case with a SMARCA4 loss of function mutation in tumor-free normal tissue, potentially representing a germline or mosaic mutation.”

Comment 13: “Finally, the authors implement the methylation-based classification in a machine learning framework. While the authors point out practical considerations for diagnostic use, it seems that the ML

algorithm makes the earlier t-SNE analysis obsolete and is “re-inventing” the classification. This seems confusing given that most of the new findings of the paper were generated with the t-SNE based classification, and it is not clear whether or how that knowledge was used in the ML framework. E.g. was the t-SNE based classification as a training input in developing the ML classifier?”

Response: We apologize for not making this aspect clear enough. As also discussed in our response to comment 3, t-SNE analysis and machine learning classification serve two different purposes: t-SNE combined with DBSCAN is used to define classes while the machine learning algorithm is used to classify novel “diagnostic” cases. The classes defined by t-SNE and DBSCAN were indeed used as training input for the machine learning classifier. To address the reviewers point we have redesigned Supplementary Figure S1 to provide a more detailed overview of the study design and how the different methods were used.

Comment 14: “The performance evaluation is somewhat hard to follow, again because it is not clear whether this implemented the earlier t-SNE classification, or what the overlap of the two classification approaches is here. E.g. were the reclassifications pointed out in lines 392f and 397f consistent with the t-SNE analysis? If not, was it expected and what are the general differences in performance one should expect with the two approaches?”

Response: The performance evaluation was solely done on the data from the machine learning classifiers. We agree with the reviewer that including the combined t-SNE plot of the reference and test set is not useful here, especially as we do not discuss it in the manuscript. Classification of new cases should solely be done using a dedicated classifier due to the limitations of t-SNE described above (response to comment 3). We see that including the combined t-SNE plot is in clear contrast to this and confusing for the reader. Therefore, we decided to remove the respective panel from Figure 5.

Reviewer 3

Comment 1: “However, because the study revolves around previously made diagnoses from various centers and uses new molecular data for correlation, the fundamental question is whether or not the reclassified tumors were actually misdiagnosed to begin with. Also, as the WHO classification for head and neck tumors was updated in 2017, how many of these samples were diagnosed before that time when modern, more updated nomenclature was not in use?”

Response: We assume that the ‘reclassified tumors’ that the reviewer refers to in the comment are the five cases from the sinonasal test that were assigned to a divergent DNA methylation class in the validation process. All of these samples have been diagnosed between 2017 and 2021, so the definition of all tumor classes – including the more recently established entities/subcategories NUT midline carcinoma and IDH2 mutated SNUCs – would have been available from the 2017 WHO classification.

Comment 2: “What is “normal sinonasal tissue” - where was it collected from, and was each specimen confirmed to be free of tumor? In how many cases was this tissue adjacent to tumor?”

Response: The exact anatomic site that the normal tissue has been derived from is annotated in Supplementary Table S3. All specimens were retrieved from independent patients undergoing sinonasal endoscopic surgery due to non-neoplastic conditions. All samples were histologically reevaluated and confirmed to be free of tumor before DNA extraction. We have added an additional paragraph to the ‘Material and Methods’ section of the updated manuscript file to address the reviewer’s point.

Comment 3: “Why were so many neuroendocrine carcinomas (11/24) and SNUCs (15/84) excluded due to noise? How does this impact the results?”

Response: We thank the reviewer for making us aware of this indeed very interesting aspect. As discussed in our response to Comment 10, possible explanations for the exclusion of cases as noise points include that these cases may correspond to hitherto unrecognized rare tumor classes or even non-sinonasal cancers e.g., tumors originally arising from the palate or brain with continuous infiltration of sinonasal structures or distant metastases from an unrecognized primary site. These tumors would be prone to be histologically classified as SNUCs due to their unusual morphology. Furthermore, expression of neuroendocrine markers is not uncommon in advanced and potentially dedifferentiated carcinomas, making the classification as neuroendocrine carcinoma more likely. As we cannot rule out

that these cases indeed correspond to distinct entities, we have included them in the publicly available dataset to facilitate their characterization in future studies. To address the reviewer’s point, we have extended our Discussion on this aspect (pp. 16-17).

Comment 4: “Were histopathologic diagnoses provided locally at each contributing center or were they re-reviewed centrally for this study?”

Response: All conventional histopathological diagnoses were taken from the original histopathological report or from the associated metadata if the samples were derived from a previous study. The cases were not reviewed or revised before inclusion. We have revised the Material and Methods section of the updated manuscript to make this clear to the reader (p. 22).

Comment 5: “The SMARCB1 altered cluster is well established as a subtype of SNUC, especially increasingly recognized in the last few years. How many tumors in this cohort were classified broadly as SNUC without consideration of this specific subtype, or before recognition of this entity?”

Response: Overall, 27 specimens fell into the SMARCB1 group, including 18 samples that were initially classified as SNUCs. In 11 cases, SMARCB1-deficiency had been recognized before inclusion of the sample in our study. From the remaining seven cases, four were diagnosed prior to 2017. As mentioned by the reviewer, SMARCB1-deficient tumors are only considered a subtype of SNUCs in the 2017 WHO classification and not a distinct entity, that’s why we decided to label these tumors as SNUCs in the t-SNE analysis.

Comment 6: “Regarding the SNUC misclassified ACC cluster - were the 25 cases from multiple centers or there was a predominance of origin from one center? To state that “tumors that represent previously misclassified adenoid cystic carcinomas are highly aggressive” may be an overstatement, as they should technically follow the natural biological behavior of ACC as opposed to SNUC.”

Response: Most of the 25 tumors from the ACC methylation class were derived from Center 2. Of note, as Center 2 is the institution of the main authors, it was also the overall biggest contributor to the reference cohort, providing 32% of all cases. The remaining cases were derived from six different institutions. A detailed table for review purposes is provided below:

	Center 2	Center 5	Center 10	Center 11	Center 13	Center 15	Other centers
All histologies	16	1	1	1	4	1	1
ACC	10	1	0	0	1	0	1
ADC	2	0	0	0	0	0	0
PDCA	0	0	0	0	3	0	0
SNUC	4	0	1	1	0	1	0

Response Letter Table 2: Samples from the ACC tumor class according to the initial histopathological diagnosis and the providing Center.

It is difficult to draw clear conclusions from this observation as most centers only provided a single case to the ACC class. However, both histomorphological ACCs and SNUCs were provided by multiple institutions, so there seems to be no apparent bias.

We agree with the author that our statement describing tumors from the ACC class as highly aggressive is not backed by the survival data. We have rephrased the Abstract accordingly (p. 3), which now reads: “This includes two classes with neuroendocrine differentiation, characterized by IDH2 or SMARCA4/ARID1A mutations with an overall favorable clinical course, one class composed of highly aggressive SMARCB1-deficient carcinomas and another class with tumors that represent previously misclassified adenoid cystic carcinomas.”

Comment 7: “The results from comparing survival based on class itself are not generalizable without more detailed patient data (e.g., tumor extent, metastases, treatments rendered), and thus should be interpreted with caution.”

Response: We agree with the reviewer’s opinion and regret that we were not able to include additional clinical factors or treatment information as this could not be retrieved by the supplying centers. We now discuss this limitation in the Discussion section of the updated manuscript (p. 21).

Comment 8: “Did the authors compare survival between SNUC ACC and the diagnosed ACC cohorts?”

Response: We thank the reviewer for bringing up this important point. In total, follow-up was available for 14 patients, including eight cases that were classified as ACC at the initial histopathological review. There were three events in each group and the median survival time was 47.4 and 39.0 months, respectively. Using the log-rank test, this difference was not significant ($p = 0.5$). We have the following sentence to the revised Results section of the manuscript (p. 13): “In the ACC methylation class, we observed no significant difference in survival between tumors that were classified as adenoid cystic carcinoma or as SNUC by conventional histopathology ($p = 0.5$).”

Comment 9: “What is the source of the 8,065 tumor and normal samples?”

Response: DNA methylation data of non-sinonasal tumor and normal tissue specimens were obtained from publically available datasets (e.g., GEO, TCGA) as well as our own previous studies. The exact source of every sample can be seen in Supplementary Table S5.

Comment 10: “Regarding the 34 cases excluded in the cluster analysis, why do the authors think they were incompatible? The comment regarding these being rare diagnoses does not apply and is an inadequate explanation, as only a minority of the noise points are singular diagnoses.”

Response: We agree with the reviewer that our initial discussion of potential reasons for the classification of 34 cases as noise points was not sufficient. In our opinion, there are several aspects that could explain this:

1. Technical variations introduced during generation of DNA methylation data as well as array quality can lead to slightly divergent profiles.
2. Rare but already recognized WHO tumor classes that do not aggregate in a separate group due to insufficient numbers of cases (as for example biphenotypic sinonasal sarcoma).
3. Potentially novel, unrecognized tumor classes also with insufficient numbers of cases to define a new class using DNA methylation data
4. Tumors of non-sinonasal origin, either with continuous growth from neighboring anatomic regions (e.g., tumors originally arising from the palate or brain with infiltration of sinonasal structures) or distant metastases from an unrecognized primary site.

We have extended our discussion on this aspect in the revised manuscript file (pp. 16-17).

Comment 11: “WHO classification requires citation.”

Response: We have added a citation for the 2017 WHO classification in the updated manuscript version (p. 5).

Comment 12: “H-score description requires a citation.”

Response: We have included a citation for the publication that originally described the implementation of the H-score for immunohistological profiling (p. 23).

Comment 13: “Please relabel Conclusions as Discussion section.”

Response: We thank the reviewers for making us aware of this error, which has been corrected (p. 16).

Comment 14: “Some of the interpretation of results which should be in the Discussion section are directly written in the Results section; this should be separated accordingly.”

Response: We have revised the manuscript when appropriate. However, several of the findings in the results section build on each other so that for a better readability we have decided to include some of the discussion aspects already in the results.

Comment 15: “Similarly, some information in the Results section should belong in the Methods section.”

Response: We have revised the manuscript when appropriate. However, we think that repeating some of the methodological aspects in the results section facilitates better readability and makes the manuscript easier to follow, as the reader does not have to go back and forth between the Results and Material and Methods section.

Comment 16: “Many statements are made without reference support or based on anecdotal evidence (“Early clinical data suggests that these patients might benefit from treatment with AURKA, EZH2 or PD1 inhibitors.”). These statements should be avoided overall given lack of scientific basis.”

Response: We carefully revised the manuscript and checked for missing references which have been added accordingly (e.g., reference 24, p. 12).

Comment 17: “Please proofread the manuscript and figures to address all typos.”

Response: The manuscript and figure legends were proofread and we hope to have addressed all errors.

Comment 18: “Parts of the manuscript are written in informal language and should more reflect formal scientific writing.”

Response: The manuscript has been proofread by native English speakers and we hope to have addressed all parts of informal language.

Reviewer 4

Comment 1: “This paper is focused on diagnostic accuracy and thus in both the methylation analysis and proteomics analysis, molecular comparisons (i.e. differential analysis followed by enrichment) were not performed across putative subtypes. Given the discovery of methylation subtypes (driven by differences in methylation sites in specific genes), understanding the methylation-specific differences between subtypes could better elucidate the biological differences between tumor subclasses. In the case of the proteomics experiment, comparisons were made of the tumor subclass vs. normal tissue. While identifying markers differing from normal tissue is important for pathology diagnosis, there is significant value in identifying the broader molecular characteristics that differ between types. This is particularly true in the subgroups (ACC and NEC-like SMARCA4/ARID1A) where a clear driver is not readily apparent.”

Response: As the reviewer states, our study's primary focus was to perform an innovative classification on a comprehensive cohort of sinonasal tumors, focusing on the diagnostic utility of this approach. While a detailed evaluation of the mechanistic differences between the identified SNUC subtypes is highly interesting, we think that this should be further investigated in a separate study. The currently selected LFQ approach for the proteomic analysis is well suited to identify potential diagnostic markers as well as patterns of potential cell of origins. A mechanistic study focusing on the activation status of signaling pathways would definitely benefit from a TMT or DIA approach that can detect more and less abundant proteins as well as PTM profiling. We are currently preparing such a study, but we think that this should not be a major focus of the current study. Furthermore, DNA methylation signatures are usually quite uninformative, as they are distributed across the whole genome and mainly located in the gene body region. The regulatory effect of DNA methylation in these regions is only poorly understood. Please also see our Response to Comment 5 by Reviewer 2.

To address the reviewer's point, we have conducted a functional proteomic analysis using the comparison between different tumor classes (Response Letter Figure 3).

Response Letter Figure 2 (new Supplementary Figure S7): Results from functional proteomic analysis.

A Tumors from the NEC-like IDH2 class were enriched for alterations in mitochondrial processes, including citric acid cycle.

B ACC class tumors showed evidence for alterations in MAPK-related signaling pathways.

C The few significant functional terms for cases from the NEC-like SMARCA4/ARID1A class were mainly associated with translational processes.

Tumors from the NEC-like IDH2 class were enriched for several functional terms related to mitochondrial processes, including citric acid cycle. This is in line with the well-known oncogenic mechanism of mutated IDH1/2, disrupting the citric acid cycle and producing the oncometabolite 2-hydroxyglutarate. ACC class tumors showed frequent evidence for alterations in MAPK-related signaling pathways, also in line with previous reports. While the few significant functional terms for cases from the NEC-like SMARCA4/ARID1A class were mainly associated with translational processes, we did not find any evidence for recurrently affected signaling pathways. This might be due to the limitations discussed above. We have added our results to the updated manuscript file (pp. 12 & 20) and have included a new Supplementary Figure S7. The limitations of our analysis are also discussed in the updated Discussion section (p. 21).

Comment 2: “The first paragraph of the results (Identification of DNA methylation ...) is not completely clear. The first tsne/DBSCAN combination determined outlier samples. Once outlier samples were removed from the cohort, was the tsne re-applied and DBSCAN used to determine the number of clusters/groups? It is not clear if the tsne/DBSCAN clustering was used from the filtering step or was rebuilt once outliers were removed. It seems that rebuilding the model excluding outliers would be more robust, so perhaps this was done.”

Response: We thank the reviewer for bringing up this important point. We performed DSCAN cluster analysis before and after removing the identified outlier samples. The optimal number of classes as well as the assignment of the non-noise cases to these classes was identical. All downstream analyses including classifier development were performed after exclusion of the outlier samples. To address this point, we have revised Supplementary Figure S1 of the updated manuscript as well as the Methods section (p. 24), which now reads: “Comparison of the number of classes and the assignment of the non-outlier cases to these classes revealed no differences before and after exclusion of the outlier samples.”

Comment 3: “Second para of results: “Iterative random down-sampling and correlation analysis” seems like it would be more understandable as “Iterative random down-sampling with correlation analysis”. That is, the two methods are tied together to assess stability of the tsne analysis (not necessarily the classes/clusters).”

Response: We revised this sentence as requested by the reviewer (p. 6).

Comment 4: “Second para: It seems that the sensitivity/specificity of methylation profiles for the non-SNUC related groups is 1/1. Is this correct? It may be worth mentioning this point explicitly. The clustering uniquely separated these other tumor types perfectly (if that is the case).”

Response: While the reviewer’s interpretation of this finding is technically correct, we would rather opt to not use the terms specificity and sensitivity here, as this is referring to the samples of the reference set. This cohort was not used for the classifier evaluation and we think that using these terms here might be confusing for the reader.

Comment 5: “The authors should be commended for examining potential batch effects. One further variable to consider could be the study (from Supplemental Table 3). It does not appear to be a confounder, but perhaps the authors have tested this variable.”

Response: We thank the reviewer for the positive and constructive feedback. We have added an additional panel E to Supplementary Figure S2 showing the t-SNE with the original study that the data was derived from annotated by color. Some entities are enriched for certain studies, as these investigations particularly focused on these tumor classes (e.g, Craniopharyngiomas from Capper et al. *Nature* 2018). For all entities that were derived from multiple studies, no apparent batch effects can be observed (e.g., NEC-like IDH2, olfactory neuroblastoma).

Comment 6: “Figure 3A: The legend shows that the data is centered log2 intensity. It is not clear if this means that the data is centered by protein. The heatmap does not appear to have centered rows as many rows are primarily one color.”

Response: Following the suggestion by Reviewer 3 and to improve consistency, the heatmap in Figure 3 has been replaced by a t-SNE plot.

Comment 7: “Figure 3B: It would be important to label these axes. Is the x axis log2 or log10 fold changes?”

Response: We thank the reviewer for making us aware of the missing axis labels. The x axis shows log₂ fold changes. The figure has been revised accordingly.

Comment 8: “A number of sections (such as the proteomics section) do not contain many quantitative results. For instance, some of the enrichment findings are described but they do not have the corresponding p values in the text. The figures provide a summary of the statistical results, but specific findings could be more precisely reported if the p values (or FDR) are included.”

Response: We have revised the Results section of our manuscript to include FDR values for the enrichment analysis mentioned in the text (pp. 11-12).

Comment 9: “Figure 5D: Including the AUC can provide a numerical summary of the curve and further emphasize the high sensitivity/specificity of the classifier. Also, it is not clear what data the ROC is derived from or what thresholds are varied to generate the ROC points.”

Response: We apologize for the insufficient description of Figure 5D. The ROC curve shows the sensitivity and specificity of the outlier detection, i.e., the binary classification ‘sinonasal’ versus ‘unknown’. Based on comment 3 by Reviewer 1, we have decided to use the terms ‘outlier detection specificity/sensitivity’ to make this clearer for the reader. Additionally, we have added the AUC to Figure 5D, enhanced the respective figure legend and added a detailed description of this analysis to the ‘Classifier evaluation’ subsection.

REVIEWERS' COMMENTS

Reviewer #1 (Remarks to the Author):

The authors have addressed my concerns.

Reviewer #2 (Remarks to the Author):

Thanks to the authors for submitting a comprehensively revised version of the manuscript. My main concern in the initial submission was the presentation of the data and the rationale for data analysis and generation of new data. The authors have improved both aspects, which will make it easier for the reader to appreciate and potentially build on the findings made in this paper. I support the publication of this revised article.

Reviewer #3 (Remarks to the Author):

Thank the authors for their thorough and thoughtful responses to all of the reviewers comments. There is certainly more clarity with regards to the methodology and interpretation of results.

However, there remains several concerns regarding the manuscript content:

- 1) "Technical variations introduced during generation of DNA methylation data as well as array quality can lead to slightly divergent profiles." This statement suggests that the methods utilized for tumor characterization are not very consistent, especially for the targeted pathology at hand, which is a major limitation.
- 2) "Tumors of non-sinonasal origin, either with continuous growth from neighboring anatomic regions (e.g., tumors originally arising from the palate or brain with infiltration of sinonasal structures) or distant metastases from an unrecognized primary site." This should really be clarified and details provided on how many such pathologies there were, as they really should be excluded from the analysis.
- 3) Lack of central histologic interpretation may introduce bias, and I would encourage the authors consider doing this to ensure that there is consensus on the analyzed pathologies. As it stands, this is a considerable limitation of the study as well.

Reviewer #4 (Remarks to the Author):

This paper represents an important study on the diagnostic potential of methylation in sinonasal tumors including the detection of "outlier" tumors that may require further study. The reviewers successfully and thoughtfully addressed my concerns, particularly on the clarity of presentation of the results. I have no further concerns.

Point-by-point response

Reviewer comments are shown in black.
Author responses are shown in blue.

Reviewer 3

Comment 1: "Technical variations introduced during generation of DNA methylation data as well as array quality can lead to slightly divergent profiles." This statement suggests that the methods utilized for tumor characterization are not very consistent, especially for the targeted pathology at hand, which is a major limitation.

Response: Array-based DNA methylation analysis is usually very robust as suggested by previous work comparing the results from independent analyses of the same samples at different laboratories (e.g. Capper et al. Nature 2018). It must be noted that technical variation is just one possible explanation for the exclusion of samples during the assembly of the reference cohort, as we further elaborated in the manuscript. Only 8 % of samples were excluded from the reference cohort, although data was obtained from more than 17 different centers. It is unknown to which degree technical variations contributed to the exclusion of these samples, but we suspect that the proportion is small.

Comment 2: "Tumors of non-sinonasal origin, either with continuous growth from neighboring anatomic regions (e.g., tumors originally arising from the palate or brain with infiltration of sinonasal structures) or distant metastases from an unrecognized primary site." This should really be clarified and details provided on how many such pathologies there were, as they really should be excluded from the analysis.

Response: The statement that some of tumors might be of non-sinonasal origin only refers to cases that were excluded during the assembly of the reference cohort as they did not correspond to a stable epigenetic class. Therefore, none of these samples were used for subsequent analysis. We clarify this in the updated version of the manuscript by adding the following sentence (p. 12): "Of note, none of these tumors were used for the development of the classifier as they did not correspond to a stable epigenetic class and were therefore excluded from further analyses."

Comment 3: Lack of central histologic interpretation may introduce bias, and I would encourage the authors consider doing this to ensure that there is consensus on the analyzed pathologies. As it stands, this is a considerable limitation of the study as well.

Response: We agree with the reviewer that this is a potential limitation and have included this in the respective paragraph of the revised manuscript file which now reads (p. 17): "Third, we did not perform central histopathological review of the cases included in this study. Therefore, the quality of the given conventional diagnoses might differ between the providing institutions due to different expertise in the diagnosis of sinonasal tumors."